# Benzoxazinoids Biosynthetic Gene Cluster Identification and Expression Analysis in Maize under Biotic and Abiotic Stresses

**DOI:** 10.3390/ijms25137460

**Published:** 2024-07-07

**Authors:** Xiaoqiang Zhao, Zhenzhen Shi, Fuqiang He, Yining Niu, Guoxiang Qi, Siqi Sun, Xin Li, Xiquan Gao

**Affiliations:** 1State Key Laboratory of Aridland Crop Science, College of Agronomy, Gansu Agricultural University, Lanzhou 730070, China; shizz@gsau.edu.cn (Z.S.); hefq6125@163.com (F.H.); niuyn@gsau.edu.cn (Y.N.); qigx1321@163.com (G.Q.); 15045240973@163.com (S.S.);; 2State Key Laboratory for Crop Genetics and Germplasm Enhancement, Nanjing Agricultural University, Nanjing 210095, China

**Keywords:** biotic/abiotic stresses, benzoxazinoids biosynthetic gene cluster, maize, gene expression, RNA-sequencing, phylogenetic analysis, gene structure

## Abstract

Benzoxazinoids (BXs) are unique bioactive metabolites with protective and allelopathic properties in maize in response to diverse stresses. The production of BXs involves the fine regulations of BXs biosynthetic gene cluster (BGC). However, little is known about whether and how the expression pattern of BGC members is impacted by biotic and abiotic stresses. Here, maize BGC was systemically investigated and 26 BGC gene members were identified on seven chromosomes, for which Bin 4.00–4.01/4.03–4.04/7.02 were the most enriched regions. All BX proteins were clearly divided into three classes and seven subclasses, and ten conserved motifs were further identified among these proteins. These proteins were localized in the subcellular compartments of chloroplast, endoplasmic reticulum, or cytoplasmic, where their catalytic activities were specifically executed. Three independent RNA-sequencing (RNA-Seq) analyses revealed that the expression profiles of the majority of BGC gene members were distinctly affected by multiple treatments, including light spectral quality, low-temperature, 24-epibrassinolide induction, and Asian corn borer infestation. Thirteen differentially expressed genes (DEGs) with high and specific expression levels were commonly detected among three RNA-Seq, as core conserved BGC members for regulating BXs biosynthesis under multiple abiotic/biotic stimulates. Moreover, the quantitative real-time PCR (qRT-PCR) verified that six core conserved genes in BGC were significantly differentially expressed in leaves of seedlings upon four treatments, which caused significant increases in 2,4-dihydroxy-7-methoxy-1,4-benzoxazin-3-one (DIMBOA) content under darkness and wound treatments, whereas a clear decrease in DIMBOA content was observed under low-temperature treatment. In conclusion, the changes in BX metabolites in maize were regulated by BGC gene members in multiple stress presences. Therefore, the identification of key genes associated with BX accumulation under biotic/abiotic stresses will provide valuable gene resources for breeding maize varieties with enhanced capability to adapt to environmental stresses.

## 1. Introduction

Benzoxazinoids (BXs) are a group of cyclic hydroxamic acids with the 1,4-benzoxazin-3-one skeleton (Figure 1), commonly known as a class of indole-derived plant chemical defense compounds and highly abundant in several gramineous species, including maize (*Zea mays* L.) [1], wheat (*Triticum aestivum* L.) [2], rye (*Secale cereale* L.) [3], and barley (*Hordeum vulgare* L.) [4]. Since their discovery in the 1950s, BXs displayed a wide range of antifeedant, insecticidal, antimicrobial, and allelopathic activities [5,6]. In maize, the well-known predominant BXs included 2,4-dihydroxy-1,4-benzoxazin-3-one (DIBOA), 2,4-dihydroxy-7-methoxy-1,4-benzoxazin-3-one (DIMBOA), as well as their glucoside compounds. DIBOA-Glc and DIMBOA-Glc could account for 0.1–0.3% of maize fresh weight (FW), especially in young seedlings [1,7,8,9]. Additionally, by an uncommon reaction involving a hydroxylation and a likely ortho-rearrangement of a methoxy group, DIMBOA-Glc was converted into the 2-(2,4,7-trihydroxy-8-methoxy-1,4-benzoxazin-3-one)-b-D-glucoside (TRIMBOA-Glc, a new BX intermediate), which was structurally similar to the 2,4,7-trihydroxy-1,4-benzoxazin-3-one-glucoside (TRIBOA-Glc) (Figure 1) [10]. Meanwhile, DIMBOA-Glc was also catalyzed to a more toxic 2-hydroxy-4,7-dimethoxy-1,4-benzoxazin-3-one glucoside (HDMBOA-Glc) by methyltransferases (Figure 1) [9]. In response to insect feeding, DIMBOA-Glc was activated by glucosidases to form DIMBOA in maize, which then decreased the in vivo endoproteinase activity in the larval midgut of the European corn borer, thus, limiting the availability of amino acids and reducing larval growth [11]. DIMBOA was also reported to influence some nervous systems, detoxication, and inactivate some hydrolysis enzymes of Asian corn borer (ACB; *Ostrinia furnacalis*; Lepidoptera, Pyralidae) larvase [12]. After 60 h infestation for the first instar larvae of *Sesamia nonagrwides* (Lepidoptera: Noctuidae), the leaves of infested maize were injured, resulting in a significant increase in DIMBOA content by 42–96% in leaves [13]. DIMBOA also induced callose accumulation to elevate maize corn leaf aphid resistance [14]. The enhanced resistance to sheath blight was related to DIMBOA accumulation in mycorrhizal maize [15]. Furthermore, root-exuded DIBOA-Glc and DIMBOA-Glc in rye that were absorbed by the roots of co-cultivated hairy vetch (*Vicia villosa* Roth) plants and translocated to their shoots [16], or DIMBOA showed a strong allelopathy in wheat by affecting the germination and growth of weeds [17].

Previous studies have shown that while BX accumulation in maize was strongly associated with the resistance to the biotic stresses described above, they were also regulated by diverse abiotic stresses, including water deficit [18,19], temperature control [19], mechanical damage [20,21], and light output (i.e., photoperiod, light intensity, light spectra, and ultraviolet radiation) [22]. The genetic difference in BX levels was caused by the main adaptive selection in stress presences, which were against various abiotic stresses [23]. In maize, BXs did not exist in seeds, while BX levels, particularly DIMBOA were increased after germination [18]. Through high-performance liquid chromatography (HPLC) analysis, the DIMBOA content of maize etiolated seedlings grown in the darkness was found to be twice that of those grown in natural light [21]. Similarly, using the liquid chromatography–tandem mass spectrometry (LC-MS/MS) method, the DIMBOA content of HKI 161 (a susceptible genotype) and HKI 193-2 (a least susceptible genotype), were 0.94 and 1.53 mg g^−1^ FW of seedlings in the darkness, whereas they were 0.68 and 1.31 mg g^−1^ FW in light condition, with a decrease of 27.7 and 14.4%, respectively [22], indicating that light may suppress BXs production, particularly the DIMBOA level in maize seedlings. However, the role of other abiotic stresses affecting the formation and degradation of BXs in different maize tissues remains unknown. In addition, the core biosynthetic pathway for DIMBOA-Glc has been clearly characterized in maize [7,9], and nine genes encoding corresponding enzymes in this pathway have been identified. These genes are located on chromosomes 1 and 4, respectively, and their biological functions, which are involved in multiple insect resistance have been studied extensively [7,9,24,25,26]. Whether there were other genes involved in BX biosynthesis, and how these genes were expressed under different biotic and abiotic stresses to affect BX biosynthesis, remain poorly understood.

Considering the above interesting ecological functions of BXs, revealing the molecular mechanisms of BXs biosynthesis and decomposition by different stimulations will be beneficial to elucidate their evolutionary adaptability and breeding programs in maize. Fortunately, the genetic variation in maize BXs accumulation is greatly facilitated by recent genomics advances, facilitating the discovery of previously unknown genes and functions of BXs biosynthesis. In the context of these considerations, the present study aims to (i) identify BXs biosynthetic gene cluster (BGC) gene members in maize, and comprehensively analyze their phylogenetic relationships, sequence features, chromosome distributions, and subcellular localization; (ii) analyze the expression patterns of all BGC gene members in various biotic/abiotic-induced tissues of maize via RNA-sequencing (RNA-Seq); (iii) verify the relative expression patterns of the core conserved BGC genes and DIMBOA accumulation in response to different biotic/abiotic stresses by quantitative real-time PCR (qRT-PCR) and HPLC. The findings may provide valuable information for further functional characterization of BGC genes and their breeding application in maize.

## 2. Results

### 2.1. Genome-Wide Identification and Chromosome Distribution of BGC Gene Members

By retrieving maize reference genome B73_V4, a total of twenty-six BGC gene members were identified through the Gene Ontology (GO) analysis, Kyoto Encyclopedia of Genes and Genomes (KEGG) analysis, and NCBI non-redundant protein sequences (NR) annotation, including two *BX1* (*Zm00001d034453* and *Zm00001d048709*), one *IGL1* (*Zm00001d034461*), one *BX2* (*Zm00001d048710*), one *BX3* (*Zm00001d048702*), one *BX4* (*Zm00001d048703*), one *BX5* (*Zm00001d048705*), three *BX6* (*Zm00001d048634*, *Zm00001d044954*, and *Zm00001d023933*), three *BX7* (*Zm00001d049179*, *Zm00001d049181*, and *Zm00001d049265*), eleven *BX8* (*Zm00001d034692, Zm00001d002753, Zm00001d005174*, *Zm00001d005456*, *Zm00001d041328*, *Zm00001d048707*, *Zm00001d019250*, *Zm00001d019254*, *Zm00001d019258*, *Zm00001d019259*, and *Zm00001d019265*), one *BX9* (*Zm00001d031209*), and one *BX13* (*Zm00001d007718*) (Figure 2A; Appendix A). These BGC gene members were distributed on seven chromosomes, except for chromosomes 5, 6, and 8 (Figure 2B; Appendix A). Meanwhile, seven BGC gene members were mainly located in Bin 4.00–4.01 (1.85~3.84 Mb) region, three BX7 gene members were mainly located in Bin 4.03–4.04 (19.41~23.64 Mb) locus, and five *BX8* gene members were mainly located in Bin 7.02 (23.98~24.77 Mb) interval (Figure 2A; Appendix A). These findings indicate that Bin 4.00–4.01, Bin 4.03–4.04, and Bin 7.02 regions are important loci regulating BXs biosynthesis.

### 2.2. Characterization and Subcellular Location of BGC Gene Members

For 26 BGC genes identified, their coding sequence (CDS) length ranged from 591 (*BX7*; *Zm00001d049265*) to 1629 bp (*BX2*, *Zm00001d048710*), and the respective numbers of amino acids (aa) varied from 261 to 542 aa (Appendix A). Correspondingly, the predicted molecular weight ranged from 28.03 to 60.49 KDa (Appendix A). In addition, the hydrophilic (GRAVY) values of 14 BX proteins (approximately 54%) were negative, indicating strong hydrophilicity (Appendix A). Meanwhile, the theoretical isoelectric point (pI) values varied from 5.15 to 9.53, indicating the range of acidity (approximately 62%) or basicity (approximately 38%) of these proteins (Appendix A). Except for *Zm00001d049179* (*BX7*), *Zm00001d049181* (*BX7-like*), *Zm00001d031209* (*BX9*), *Zm00001d048707* (*BX8*), and *Zm00001d019265* (*BX8-like*), the instability index of other 21 BXs proteins was greater than 40 (Appendix A), suggesting their unstable nature [27]. Moreover, subcellular localization predictions showed that two BX1, one IGL1, one BX2, one BX3, and one BX5 proteins were localized in the chloroplast, implying that the BXs biosynthesis in maize was initiated by the conversion of indole-3-glycerol phosphate to indole in the chloroplast. Moreover, one BX4 protein (i.e., cytochrome P450 71C1) was localized in the endoplasmic reticulum; one BX6, three BX7, three BX13, and three BX8 proteins were localized in the cytoplasmic, while another eight BX8 and one BX9 proteins were localized in the chloroplast (Appendix A). Therefore, these identified BX members are highly diverse in maize.

### 2.3. Sequence Analysis and Classification of BGC Gene Members

To investigate the phylogenetic relationships among all BGC gene members in maize, a neighbor-joining (NJ) phylogenetic tree was generated with the 26 BX proteins identified in maize (Figure 3A). The phylogenetic analysis revealed three distinct classes, which were strongly supported by a high-confidence bootstrap value greater than 95%, designated as Class I, Class II, and Class III. Class I was further classed into two clusters, i.e., Class I-1 (including one BX9 and seven BX8 cluster proteins) and Class I-2 (including four BX8 cluster proteins); Class II included three clusters, i.e., Class II-1 (including one IGL1 and two BX1 cluster proteins), Class II-2 (including BX2-BX5 four proteins), and Class II-3 (including one BX13 and three BX6 cluster proteins); Class III consisted of two clusters, i.e., Class III-1 (including two BX7 cluster proteins) and Class III-2 (including one BX7 cluster protein). Similar findings had been reported in previous studies [28], indicating the conservation of each BGC cluster in their sequence similarities, phylogeny, and genomic physical positions.

To further explore the structural and functional characteristics of BGC genes in maize, the intron–exon structure analysis was conducted (Figure 3B). BGC gene members varied the number of exons from one to twelve, of which two *BX6* genes, i.e., *Zm00001d023933* and *Zm00001d048634* had only one exon, while the BX9 gene, i.e., *Zm00001d048709* had twelve exons (Figure 3B). Meanwhile, the first intron of *Zm00001d002753* (*BX8-like*) was the longest among all sequences (Figure 3B). Moreover, the conserved motifs of these BX proteins analyzed using the online tool MEME showed that a total of ten motifs were identified among these proteins, and the corresponding proteins belonging to the same cluster generally contain the same type and number of motifs, and their distribution patterns are similar (Figure 3C,D). Interestingly, some motifs existed only in a specific class, such as motif 1, motif 3, motif 5, and motif 7, which only appeared in Class I-1 and Class I-2; motif 10 only existed in Class II-1; and motif 8 only arose in the Class II-2 (Figure 3C).

### 2.4. Expression Patterns of BGC Gene Members under Multiple Biotic/Abiotic Stresses

BXs are a class of protective and allelopathic plant secondary metabolites that are prevalent in maize when subjected to various stresses through the entire growth period [29]. To understand the impact of different biotic/abiotic stresses on expression levels of BGC members, different organ samples from different maize genotypes treated with different types of treatments were harvested and subjected to RNA-Seq.

In this study, the elite Zheng58 seeds pre-germinated in darkness for five days, and then cultured in red, blue, and white light, and darkness for five days, respectively. Next the coleoptiles and mesocotyls under these four light spectral quality treatments, i.e., COL.Red (coleoptiles in red light), COL.Blue (coleoptiles in blue light), COL.White (coleoptiles in white light), COL.Dark (coleoptiles in darkness), MES.Red (mesocotyls in red light), MES.Blue (mesocotyls in blue light), MES.White (mesocotyls in white light), and MES.Dark (mesocotyls in darkness), with three biological replicates, a total of 24 samples were used for RNA-Seq. Intriguingly, the results showed that in comparison with darkness, the red, blue, and white light down-regulated differentially the expression of most BGC members in Zheng58 coleoptiles and mesocotyls, which were more sensitive to white light (Figure 4). Two *BX8*, *Zm00001d005174,* and *Zm00001d005456* displayed a significant up-regulation in MES.Red_vs_MES.Dark (2.33- and 2.07-fold) and MES.Blue_vs_ MES.Dark (2.62- and 1.43-fold), while they showed a significant down-regulation in MES.Dark_vs_MES.White (−3.27 and −1.63-fold). Similarly, *Zm00001d005456* expression showed a 1.30-fold increase in COL.Red_vs_COL.Dark and −1.64-fold decrease in COL.Dark_vs_COL.White, *Zm00001d007718* (*BX13*) exhibited a 1.52-fold increase in COL.Blue_vs_COL.Dark and divergent decreases in MES.Dark_vs_MES.White (−2.24-fold), COL.Blue_vs_COL.White (−1.46-fold), and COL.Dark_vs_COL.White (−3.04-fold), respectively. This result is in line with the quantification of DIMBOA contents in multiple maize seedlings growing in light and darkness conditions [22]. In addition, 15 of 26 BGC gene members also showed tissue expression specificity in white light and darkness, preferentially expressed in the mesocotyls in white light (varying from −1.78- to −7.90-fold in MES.White_vs_COL.White) or darkness (ranging from −1.02- to −4.24-fold in MES.Dark_ vs_COL.Dark), compared to coleoptiles (Figure 4). These data collectively implied that the tissue-specific expression profiles triggered by light were likely associated with DIMBOA accumulation in maize.

In the present study, the Ji853 (chilling-sensitive) and N192 (chilling-tolerant) seedlings at the V3 stage were cultured for seven days with or without 2 μM exogenous EBR at 15 and 25 °C environments. Next, the young leaves of both maize seedlings under four treatments, i.e., JCKL (leaves of Ji853 with 0 μM EBR at 25 °C), JCKEL (leaves of Ji853 with 2 μM EBR at 25 °C), JLTL (leaves of Ji853 with 0 μM EBR at 15 °C), JLTEL (leaves of Ji853 with 2 μM EBR at 15 °C), NCKL (leaves of N192 with 0 μM EBR at 25 °C), NCKEL (leaves of N192 with 2 μM EBR at 25 °C), NLTL (leaves of N192 with 0 μM EBR at 15 °C), NLTEL (leaves of N192 with 2 μM EBR at 15 °C), with three biological replicates, totally 24 samples were used for RNA-Seq. In response to low-temperature treatment, overall, the expression levels of most BGC gene members in young leaves of both Ji853 and N192 seedlings decreased at 15 °C suboptimal temperature relative to 25 °C normal temperature. Specifically, *Zm00001d007718* (*BX13*) was commonly down-regulated in JCKL_vs_JLTL (−2.07-fold) and NCKL_vs_NLTL (−2.98-fold) (Figure 5). This is in agreement with a previous study in rye, which reported that seven weeks of 4 °C cold stress caused a decrease in multiple BXs concentration and expression levels of related genes compared with untreated seedlings [30]. Moreover, the exogenous 2 μM EBR significantly positively induced the expression of *Zm00001d048710* (*BX2*) in both JCKL_vs_JCKEL (3.46-fold) and JLTL_vs_JLTEL (3.17-fold), while *Zm00001d048709* (*BX1*, 1.42-fold), *Zm00001d0048710* (*BX2*, 3.17-fold), *Zm00001d048702* (*BX3*, 1.04-fold), *Zm00001d048703* (*BX4*, 1.45-fold), *Zm00001d048705* (*BX5*, 1.36-fold), and *Zm00001d048634* (*BX6*, 2.24-fold) were positively expressed only in JLTL_vs_JLTEL (Figure 5), indicating that exogenous EBR induction could promote DIBOA formation in young leaves of chilling-sensitive maize at 15 and 25 °C conditions. Like the young leaves of Ji853 in JLTL_vs_JLTEL, the same trends of gene expression changes appeared in the young leaves of N192 in NLTL_vs_NLTEL, except for *Zm00001d048703* (Figure 5). In contrast, the 2 μM exogenous EBR significantly inhibited five gene expressions in the young leaves of N192 seedlings, namely, *Zm00001d034453* (*BX1-like*, −1.99-fold), *Zm00001d034461* (*IGL1*, −1.45-fold), *Zm00001d048634* (*BX6*, −1.57-fold), *Zm00001d005174* (*BX8-like*, −3.40-fold), and *Zm00001d007718* (*BX13*, −2.00-fold) in NCKL_vs_NCKEL (Figure 5), suggesting that exogenous EBR might not be beneficial to DIBOA-Glc and TRIBOA-Glc biosynthesis in young leaves of chilling-tolerant maize at 25 °C condition. These findings thus also reflected the complexity of exogenous EBR mediating BXs biosynthesis in different chilling tolerant maize genotypes under cold stress and unstressed environments.

In the current study, the whorl leaves of F1227 (moderate insect-resistant) and M0800 (insect-susceptible) plants to feeding and non-feeding conditions by ACB at the VT stage for 24 h. Next, the whorl leaves of both maize plants under two treatments, i.e., CKF (whorl leaves of F1227 to non-feeding by ACB), CBF (whorl leaves of F1227 to feeding by ACB), CKM (whorl leaves of M0800 to non-feeding by ACB), and CBM (whorl leaves of M0800 to feeding by ACB), with three biological replicates, a total of 12 samples were used for RNA-Seq. In terms of biotic stresses, after being fed by ACB on the whorl leaves of F1227 and M0800 plants, *Zm00001d048709* (*BX1*, 7.48-fold), *Zm00001d048710* (*BX2*, 5.52-fold), *Zm00001d048702* (*BX3*, 3.83-fold), *Zm00001d048703* (*BX4*, 2.50-fold), *Zm00001d048705* (*BX5*, 3.08-fold), *Zm00001d007718* (*BX13*, 7.53-fold), *Zm00001d048634* (*BX6*, 1.21-fold), *Zm00001d049179* (*BX7*, 2.91-fold), and *Zm00001d049265* (*BX7*, 3.49-fold) showed significant up-regulation in CKF_vs_CBF (Figure 6); likewise in CKM_vs_CBM, a few genes, i.e., *Zm00001d034453* (*BX1-like*, 1.95-fold), *Zm00001d0344613* (*IGL1*, 2.39-fold), *Zm00001d023933* (*BX6*, 6.21-fold), and *Zm00001d019250* (*BX8-like*, 2.86-fold) had significant up-regulation (Figure 6). A previous study reported that the contents of DIMBOA, DIBOA, and 2-hydroxy-7-methoxy-1,4(2H)-benzoxazin-3-one (HMBOA) in maize increased upon ACB larvae feeding on the second leaf for 48 h by HPLC [31]. Therefore, these results demonstrated that ACB leaf-feeding resistance was positively correlated to BX compound accumulation. When ACB fed on the leaves of the insect-resistant maize genotype, more BGC gene members could be activated along with the production of higher amounts of insect toxic compounds, such as DIBOA, DIBOA-Glc, TRIBOA-Glc, DIMBOA-Glc, and DIMBOA, to confer resistance to leaf-feeding by ACB.

### 2.5. Identification of Core Conserved BGC Gene Members in Maize

According to the fragments per kilobase of transcript per million mapped read (FPKM) values of 26 identified BGC gene members by the three independent RNA-Seq above, their transcripts per million (TPM) values based on the normalized scale method were calculated (Appendix A), respectively. Combining the differentially expressed genes (DEGs) (screening criteria were |log2 FC| > 1 and *p*-value < 0.05 in single comparison) of corresponding BGC gene members among all comparisons, a total of 18, 18, and 16 common BGC gene members were then identified in three independent RNA-Seq datasets, respectively. Intriguingly, 13 core conserved BGC gene members were obtained from these three sets of common DEGs in BGC from three RNA-Seq (Figure 7). These core conserved DEGs in BGC are likely associated with BX accumulation among various biotic and abiotic stresses.

### 2.6. Effect of Multiple Stress Presences on Expression Levels of Core Conserved DEGs in BGC Genes and DIMBOA Accumulation

To validate the expression patterns of core conserved DEGs in BGC, six core conserved DEGs in BGC gene members were randomly selected for qRT-PCR, and the corresponding DIMBOA content was quantified in young leaves of 14-day-old F1227 seedlings in response to control, light, temperature, and wound (mimicking ACB feeding symptom). The qRT-PCR analysis showed that these core conserved BGC gene members were significantly activated under both darkness and wound treatments, whereas they were significantly decreased under low-temperature treatment, compared to the relative controls (Figure 8A). It also noted that the relative expression levels of *Zm00001d048710* (*BX2*), *Zm00001d048705* (*BX5*), and *Zm00001d049179* (*BX7*) were higher under darkness treatment than that by wound treatment. With a noticeably different trend, *Zm00001d048702* (*BX3*) and *Zm00001d048634* (*BX6*) had a higher expression under wound treatment (Figure 8A). Additionally, the relative expression levels of *Zm00001d048703* (*BX4*) did not show a difference under both darkness and wound treatments (Figure 8A). Surprisingly, DIMBOA content significantly increased by 8.3 and 4.0% under darkness and wound treatments, compared to control treatment in the young leaves of F1227 seedlings, respectively (Figure 8B). However, low-temperature-treated leaves showed the opposite trend, i.e., with a significant decrease (46.8%) in DIMBOA level (Figure 8B). Further, Pearson correlation analysis showed that the expression levels of the six core conserved BGC gene members had a significant (*p* < 0.05) positive correlation to DIMBOA accumulation in the young leaves of 14-day-old F1227 seedlings in response to four treatments (Figure 8C). Therefore, light, temperature, and wound significantly impacted the expression of corresponding core conserved BGC gene members, suggesting that they might play a role in response to these stresses, via the BXs accumulation, especially obvious changes in DIMBOA content.

## 3. Discussion

The toxic compounds, generally arising from secondary metabolism, are an important part of plant defense, which can effectively help plants resist external stresses [32,33]. Known for their antibiotic properties, BXs were well described as natural pesticides and allelochemicals, as a typical class of secondary metabolite acting in multiple defensive roles [5,6]. Currently, studies on BXs represented by DIMBOA in maize have mainly focused on extraction [34,35], biosynthesis [36,37], and resistance to microbial and herbivory [20,29,38]. For the BXs biosynthesis in maize, the indole-3-glycerol-phosphate lyase (BX1 and IGL1) and tryptophan synthase (TSA) are capable of converting indole-3-glycerol phosphate (IGP) to indole [28]; subsequently, four cytochrome P450-dependent monooxygenases (BX2, BX3, BX4, and BX5) are involved in DIBOA biosynthesis [39,40]; DIBOA-Glc is then synthesized by glucosylation (UDP-glucosyltransferases, BX8 and BX9) of DIBOA [41]. DIBOA-Glc is the substrate of the dioxygenase (BX6) and the produced TRIBOA-Glc is metabolized by the methyltransferase (BX7) to form DIMBOA-Glc [37]. The final reaction steps leading to HDMBOA-Glc and TRIMBOA-Glc are also elucidated in maize by Zhou et al. [1].

Based on the full pathway of DIMBOA-Glc and TRIMBOA-Glc from IGP, using genome-wide identification, we found 26 BGC gene members in maize, which were mainly distributed on chromosomes 1, 2, 3, 4, 7, 9, and 10 (Figure 2A), among which, seven, five, and three BGC gene members were mainly located in the Bin 4.00–4.01 (1.85~3.84 Mb), Bin 7.02 (23.98~24.77 Mb), and Bin 4.03–4.04 (19.41~23.64 Mb) regions, respectively (Figure 2B). This result is well supported by the previous findings using genetic analysis [36,37,39,40]. Since several BGC gene members were co-localized to the same chromosomal regions, especially Bin 4.01–4.02, 4.03–4.04, and Bin 7.02, they might be co-linked genetically to determine the formation and detoxification of BXs metabolites. Early evolution relationships had revealed that BGC gene members probably originated from gene duplication and chromosomal translocation of native homologs of *BX* genes, whereas *BX6* class in maize appeared to have been under less constrained selection compared to other BGC gene members during the evolution of Panicoideae [28]. In this study, the phylogenetic tree of 26 identified BGC gene members was clearly separated into three clades and seven subclasses (Figure 3A). Namely, Class I-1 and Class I-2 were located on clade I, including *BX8* and *BX9* (a maize-specific duplicate of *BX8*) cluster (Figure 3A). This evolution might be related to the two glucosyltransferases (BX8 and BX9) displaying a very high degree of substrate specificity [42]. Interestingly, two *BX1* and one *IGL1* formed a momoclade (Class II-1) (Figure 3A), implying that they were paralogs of the *BX1* cluster in maize, which were native and extraordinarily conserved. There were thus two conserved motifs (motif9 and motif10) of three genes in the ClassII-1 cluster from the present study (Figure 3C). Moreover, the constructed phylogeny of *BX1* homologs across the grass family also showed that the *BX1* cluster originated from the duplication of *tryptophan synthase A homolog 1* (*TSAh1*) [28]. In brief, we also found that the *BX2*~*BX5* genes showed high genomic synteny among Class II-2, and their topologies were similar to those observed in the *BX1* cluster (Class II-1) phylogeny (Figure 3A). Similarly, the phylogeny of *BX6* and *BX13* (a maize-specific duplicate of *BX6*) gene members were nested within the Class II-3, which showed a pattern similar to that of the BX1 cluster (Figure 3A). These findings thus indicated that the corresponding BGC gene members in the same clade might be derived from a single origin. Notably, the three *BX7* gene members were nested in a well-defined specific clade III (including Class III-1 and Class III-2) (Figure 3A). Thereby, the limited homologs of the BX7 cluster could be identified in maize. Similar to the different *BX* clusters described in rice [43], the 26 identified BX proteins in maize harboring similar conserved motifs were found to group according to their NJ phylogenetic clustering in this study (Figure 3C). Additionally, all BXs proteins in maize were found to be strategically localized in subcellular compartments, where their catalytic activities were executed; for instance, the proteins of BX1, IGL1, BX2, BX3, and BX5 were localized in the chloroplasts (Appendix A), where they encoded the production of indole and DIBOA; like wheat [5], maize BX4 protein was localized in the endoplasmic reticulum (Appendix A) responsible for HBOA biosynthesis. Interestingly, all BX6, BX7, and BX13 were identified as cytoplasmic proteins (Appendix A), where they were involved in the hydroxylation of DIBOA-Glc, TRIBOA-Glc, and DIMBOA-Glc, respectively. Unlike other BXs proteins, UDP-glucosyltransferases (BX8 and BX9) proteins were localized in the chloroplasts (nine) and cytoplasts (three) (Appendix A), where they attached a glucose moiety to DIBOA to produce DIBOA-Glc [42].

Increasing evidence has proven that a variety of factors affected BX biosynthesis [18,19,20,21,22,29]. Light and temperature played a key role in the seedling emergence and morphology establishment in maize. Meanwhile, the newly hatched ACB larvae primarily fed on young leaves of maize during this development stage [44]. Therefore, a comprehensive understanding of the impacts of light and temperature environments and ACB infestation is important for BX biosynthesis in maize. In the current study, we examined the impact of light spectral quality, low-temperature, exogenous EBR phytohormone, and ACB attack on the expression patterns of 26 BGC gene members by three independent RNA-Seq experiments. The transcriptome analysis revealed that red, blue, and white light mediated the down-regulation expression of 26 BGC gene members in both coleoptiles and mesocotyls of Zheng58 seedlings ten days after germination (Figure 4). However, the results also showed that the expression specificity of these BGC gene members clearly depended on the different light spectral quality and tissue changes in maize, for instance, *Zm00001d048710* (*BX2*) had a significant up-regulation in MES.Red_vs_COL.Dark (1.18-fold), while it showed a significant down-regulation in COL.Red_vs_COL.White (−4.03-fold) (Figure 4). As expected, Chandra et al. [22] reported that light played a significant role in the inhibition of DIMBOA formation in maize seedlings as compared to that treated with darkness. The same phenomenon was also observed in wheat [45]. Under prolonged 4 °C low-temperature stress, Bakera et al. [30] reported that the BX synthesis-related genes in rye leaves, including *ScBX2*, *ScBX3*, and *ScBX5*, were expressed at lower levels, resulting in decreased HBOA and DIMBOA concentrations. Similarly, most BGC gene members showed negative expression levels in young leaves of both chilling-tolerant N192 and chilling-sensitive Ji853 seedlings at 15 °C low-temperature stress, yet in a strong genotype-dependent manner. For instance, *Zm00001d007718* (*BX13*) was down-regulated significantly in JCKL_vs_JLTL (−2.07-fold) and NCKL_vs_NLTL (−2.98-fold) (Figure 5). Previous studies found that DIMBOA content increased in maize after jasmonic acid (JA) treatment [46], however, the DIMBOA-Glc accumulation decreased in wheat and Job’s tears (*Coix lacryma-jobi*) [47]. Intriguingly, in this study, we found that the expression levels of the 26 BGC gene members were also upregulated by 2 μM EBR induction in maize young leaves at 25 and 15 °C environments (Figure 5), suggesting that the formation of BXs metabolites could be induced by several phytohormones, such as JA and EBR. It was also generally accepted that DIMBOA had an antifeedant effect on ACB and math, DIMBOA increased the time that the larvae required to reach the pupal stage [10,48,49]. In the present study, our results showed that the majority of BGC gene members were positively regulated in whorl leaves of insect-resistant F1227 and insect-susceptible M0800 under ACB feeding treatment at the VT stage (Figure 6). Thereby, DIMBOA might contribute to maize defense against insect feeding through the activation of its biosynthetic genes.

Fortunately, we finally identified 13 core conserved BGC gene members in this study (Figure 7). Subsequently, the significantly differential expression of six core conserved genes was verified in young leaves of F1227 seedlings across four treatments of control, darkness, low-temperature, and wound (mimicking ACB feeding symptom) (Figure 8A). At the same time, the inducement effects on DIMBOA accumulation in F1227 young leaves under different treatments were obvious (*p* < 0.05), and the DIMBOA content followed the order: Darkness > wound > control > low-temperature (Figure 8B). At the same time, our results also confirmed that the expression levels of these core conserved BGC gene members were positively and significantly correlated to DIMBOA content under four stress environments (Figure 8C), which supports the viewpoint of Niu et al. [29] very well. Therefore, the identification of core conserved BGC gene members, that were highly and specifically expressed among genotypes, developmental stages, tissues, and environmental stimulates, could provide reliable gene resources for maize breeding.

## 4. Materials and Methods

### 4.1. Genome-Wide Identification and Mapping of BGC Gene Members in Maize

BGC genes members were retrieved according to the analyses of GO (http://bioinfo.cau.edu.cn/agriGO/; accessed on 16 April 2024), KEGG (http://www.genome.jp/kegg/; accessed on 16 April 2024), and NR annotation (http://ncbi.nlm.nih.gov/; accessed on 16 April 2024) of maize reference genome B73_V4 (http://plants.ensemblgenomes.org/pub/plants/release-6/fasta/zea_mays/dna/; accessed on 16 April 2024). These BGC gene members in the physical map were mapped using the MapInspect 1.0 software (https://mapinspect.software.informer.com; accessed on 16 April 2024).

### 4.2. Sequence Analysis, Structural Characterization, Subcellular Localization, and Phylogenetic Tree of BGC Gene Members in Maize

The gDNA, CDS, and protein sequence of 26 BGC gene members in maize were obtained from NCBI public database (https://www.ncbi.nlm.nih.gov/; accessed on 19 April 2024). The exon–intron organizations of all BGC gene members were graphically displayed by the Gene Structure Display Server (GSDS2.0, http://gsds.cbi.pku.edu.cn/; accessed on 20 April 2024) [50]. The amino acids number, molecular weight, theoretical pI, GRAVY, instability index, and aliphatic index of these proteins were analyzed using the ExPASy (https://web.expasy.org/protparam/; accessed on 19 April 2024) [27]. Then the conserved motifs of these proteins were predicted using the MEME Suite program v.5.5.0 (http://alternate.meme-suite.org/tools/meme; accessed on 20 April 2024) [51], with the following parameters: the maximum number of motifs was set to 10, optimal motif width of 6 to 50 amino acid residues, and any number of repeats [51]. According to the amino acids sequences of these proteins, an NJ phylogenetic tree was built using pairwise deletion option in the MEGA 6.0 software (https://www.megasoftware.net/; accessed on 20 April 2024) [52], with a bootstrap test with 1000 iterations, and subcellular localization was also predicted using the Plant-mPLOC program (http://www.csbio.sjtu.edu.cn/bioinf/plant-multi/; accessed on 20 April 2024) [53].

### 4.3. Expression Patterns of BGC Gene Members in Different Abiotic/Biotic-Induced Tissues by RNA-Seq

The 20 soaked seeds of Zheng58 (a representative inbred line in China, that is the female parent of Zhengdan958 hybrid [54]) with double-distilled water (ddH_2_O), were pre-cultured using germinating boxes in darkness at 22 ± 0.5 °C for five days. They were subsequently placed into plant chambers and were illuminated with lamps consisting of three light-emitting diode (LED) bars specifically designed to provide a custom spectrum, including white light (photosynthetic photon flux density (PFD) 17 μM m^−2^ s^−1^, 12 h photoperiod), red light (peak wavelength 660 nm, PFD 22 μM m^−2^ s^−1^, 12 h photoperiod), and blue light (peak wavelength 450 nm, PFD 13 μM m^−2^ s^−1^, 12 h photoperiod) in each chamber, respectively. The control remained in the darkness. During culture, 20 mL of Hoagland solution was watered to each box at 2-day intervals, the temperature was set to 22 ± 0.5 °C, and relative humidity was set to 70% to ensure adequate moisture. Then the coleoptiles and mesocotyls of Zheng58 seedlings were under four treatments (darkness, white light, red light, and blue light), with three biological replicates, in total 24 samples were collected and used for RNA-Seq [55]. The published raw data of this RNA-Seq were deposited to NCBI BioProject database (accession number PRJNA917698).

The 20 soaked seeds of Ji853 (chilling-sensitive) [56] and N192 (chilling-tolerant) with ddH_2_O, were sown in plastic boxes, and cultured in a growth chamber (25 ± 0.5 °C, 300 μM m^−2^ s^−1^ white light, 12 h photoperiod). At V3 stage of seedlings, they were subsequently placed into 25 ± 0.5 and 15 ± 0.5 °C environments for 7 days, along with 5 mL of 2 μM EBR (CAS: 78821-43-9) spayed on seedling leaves each day. The control seedling leaves received same volume of ddH_2_O. A 30 mL amount of Hoagland nutrient solution was added to plastic boxes at 2-day intervals during seedling growth period. Overall, the young leaves of both maize seedlings under four treatments (0 μM EBR at 25 °C, 2 μM EBR at 25 °C, 0 μM EBR at 15 °C, and 2 μM EBR at 15 °C), with three biological replicates, in total 24 samples were collected and used for RNA-Seq. The new raw data of this RNA-Seq were deposited to NCBI BioProject database (accession number PRJNA1096968).

The seeds of F1227 (insect-resistant) and M0800 (insect-susceptible) [29] were planted at a density of 67,500 plants/hm^2^ with 0.4 m row spacing, on 22 April 2023, at the Longxi experimental stations, Gansu, China (34°58′ N, 104°23′ E, 2074 m altitude; sandy loam), with a split block design. A plastic film (0.01 mm thick, 120 cm wide) was laid out by hand over the fields and covered the soil surface, other managements unified as field. Weather data at the site in all maize life stages were recorded (Appendix A). When the plants were grown to the VT stage, the six same instar larvae of ACB were placed in the whorl leaves of corresponding plants and were allowed to feed freely for 24 h. The corresponding leaves were protected with nets to avoid ACB dispersion. For the control plants, there was no ACB in the nets. Next, the whorl leaves of F1227 and M0800 plants were under two treatments (non-feeding and feeding by ACB), with three biological replicates, in total 12 samples were used for RNA-Seq. The new raw data of this RNA-Seq were deposited to NCBI BioProject database (accession number PRJNA1121054).

After filtering, using the HISAT v.2.2.1 (https://daehwankimlab.github.io/hisat2/; accessed on 18 December 2023), the clean sequence reads of above three RNA-Seq were aligned to the *Zea mays* B73_V4 reference genome, respectively. Subsequently, the FPKM values were used to estimate the gene expression of 26 BGC gene members via the Cufflinks v.2.2.1 [57] and were visualized using the TBtools v2.030 software (https://github.com/CJ-Chen/TBtools/releases; accessed on 25 December 2023). 

### 4.4. Core Conserved BGC Gene Members Identification

For each independent RNA-Seq, the TPM values of these BGC gene members based on the normalized scale method were calculated [58]. Meanwhile, their significantly differential expression in all comparisons was estimated using a *p*-value < 0.05 and |log2 FC| > 1 criterion. Firstly, the common BGC gene members were identified that showed both TPM value > 2 in single sample and |log2 FC| > 1 and *p*-value < 0.05 in single comparison in an independent RNA-Seq. Then the core conserved BGC gene members with high, specific, and significant expression were identified among three common BGC gene members of all RNA-Seq by the VENNY 2.1 (https://bioinfogp.cnb.csic.es/tools/venny/index.html; accessed on 29 December 2023).

### 4.5. qRT-PCR Analysis of Core Conserved BGC Gene Members

The 20 soaked seeds of F1227 with ddH_2_O were sown in plastic boxes, and cultured for two weeks under four treatments in corresponding growth chambers, including control treatment (CK; seedlings growing in normal environment; 22 ± 0.5 °C, 300 μM m^−2^ s^−1^ white light, 12 h photoperiod), darkness treatment (dark; etiolated seedlings growing in darkness, 22 ± 0.5 °C), low-temperature treatment (LT; seedlings growing at 22 ± 0.5 °C for seven days and then culturing at 10 ± 0.5 °C for seven days; 300 μM m^−2^ s^−1^ white light, 12 h photoperiod), and wound treatment (wound; three leaf wounds of 3 × 3 mm^2^ were made with a scalpel on thirteen-day old seedling for 24 h [59]; 22 ± 0.5 °C, 300 μM m^−2^ s^−1^ white light, 12 h photoperiod).

Next, the total RNA was extracted with TRIZOL reagent (TIANGEN, Beijing, China) of young leaves of F1227 seedlings under four above treatments, and then 0.5 μg RNA was reverse-transcribed to produce first-strand cDNA using PrimeScript^TM^ 1st strand cDNA Synthesis Kit (TaKaRa, Japan). The Primers of six random core conserved BGC gene members were designed and synthetized [10,20] (Appendix A). Subsequently, the qRT-PCR analysis was performed using LightCycler480II fluorescent quantitative PCR instrument (Roche, Germany). The relative expression level with three biological replicates was calculated by the 2^−ΔΔCT^ method, with *Zm00001d010159* (*Actin 1*) as an internal reference gene [60].

### 4.6. Quantification of DIMBOA Contents

The DIMBOA content in young leaves of F1227 seedlings under above treatments was determined according to our previous report [29]. The freeze-dried leaves (0.2 g) were homogenized into screw-capped 10 mL centrifuge tubes and 5 mL methanol–methanoic acid solution (0.01%, *v*/*v*) was added to corresponding tube. The tubes were rotated and placed in darkness for 12 h. and then centrifuged at 12,000 rpm (Centrifuge 5425/5425 R; Eppendorf, Germany) for 20 min at 4 °C. The 600 μL supernatants were then slowly passed filtration column for HPLC-MS analysis. DIMBOA Standard (CAS: 15893-52-4) was used to optimize the mass spectrometric parameters and fragment spectra.

### 4.7. Statistical Analyses

For the qRT-PCR expression levels of six core conserved BGC gene members and DIMBOA content in young leaves of F1227 seedlings under above treatments, their ANOVA was performed using the IBM-SPSS Statistics v.19.0 (https://www.Ibm.com/products/spss-statistics, accessed on 23 May 2024), and their interactive ring Pearson correlation diagram was drawn using the Genescloud tool (https://www.genescloud.cn, accessed on 24 May 2024).

## 5. Conclusions

In conclusion, we totally identified 26 BGC gene members on seven chromosomes of maize, 15 of which were mainly concentrated in Bin 4.00–4.01/4.03–4.04/7.02 regions. The proteins of these BGC gene members were grouped into three clades and seven subclasses by phylogenetic clustering, with each BXs cluster harboring similar conserved motifs, and all corresponding proteins were exclusively localized in the chloroplast, endoplasmic reticulum, or cytoplasmic, where their catalytic activities were executed to form different BXs metabolites. Moreover, the expression profiles of all BGC gene members were affected by multiple treatments with light spectral quality, low-temperature, EBR induction, as well as ACB infestation using three independent RNA-Seq analyses, and 13 core conserved BGC gene members were further identified that showed high and specific expressions. Additionally, six randomly selected core conserved BGC gene members displayed significantly differential expression levels in maize young leaves under control, darkness, low-temperature, and wound treatments, resulting in significant increases in DIMBOA accumulation under darkness and wound treatments, whereas a dramatical decrease in DIMBOA level under low-temperature stress. Therefore, these findings may lay the foundation for further studies to elucidate the functions of BGC gene members in maize and the identified core conserved BGC gene members could be potential candidate genes for maize breeding.

## Figures and Tables

**Figure 1 ijms-25-07460-f001:**
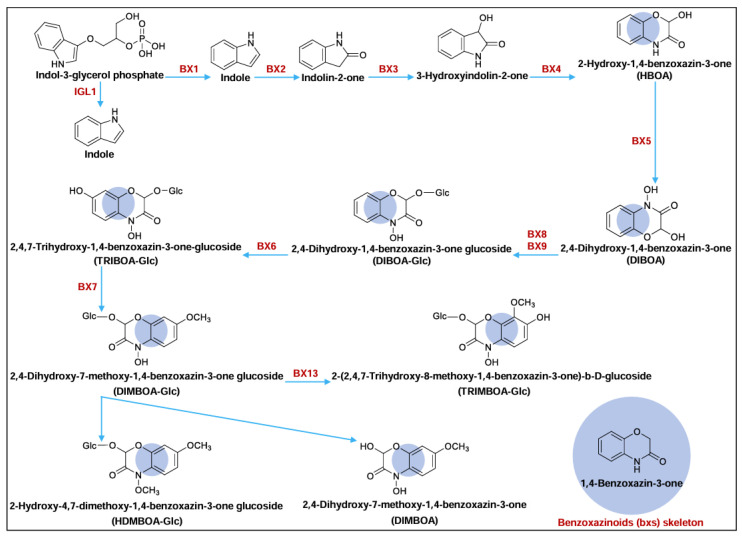
The skeleton structure of benzoxazinoids (BXs), major BXs, and corresponding enzymes, including IGL1, BX1, BX2, BX3, BX4, BX5, BX6, BX7, BX8, BX9, and BX13, catalyzing their biosynthesis in maize.

**Figure 2 ijms-25-07460-f002:**
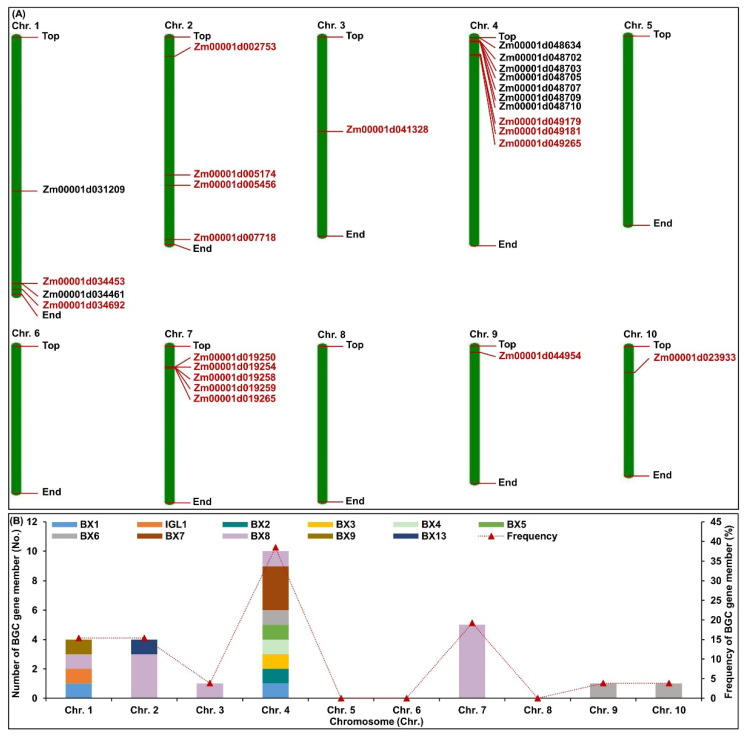
(**A**) Chromosomal locations of 26 benzoxazinoids (BXs) biosynthetic gene cluster (BGC) gene members detected in maize. Nine BGC gene members (black) were identified in previous study, and seventeen BGC gene members (red) were identified in present study. (**B**) The summary of 26 BGC gene members on ten chromosomes.

**Figure 3 ijms-25-07460-f003:**
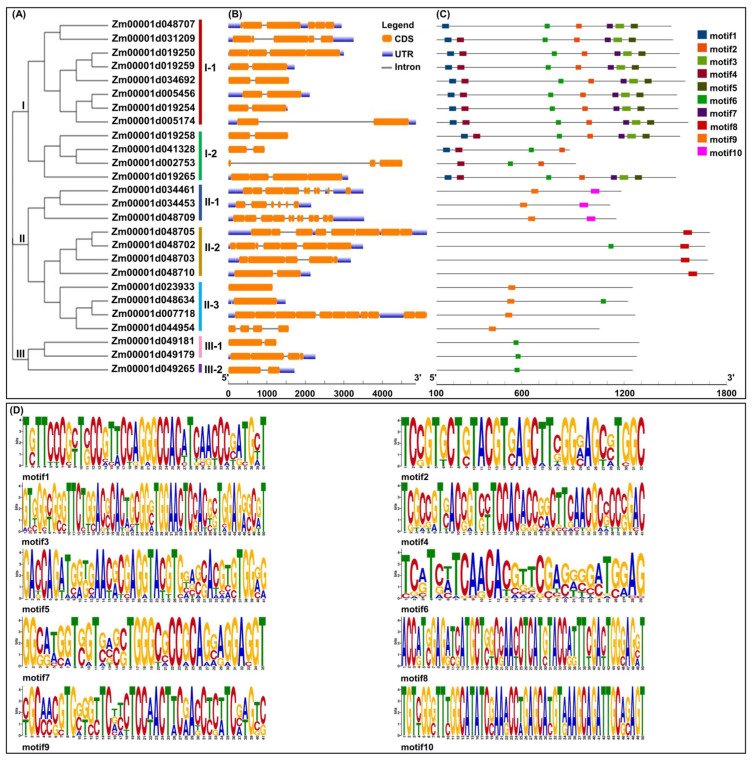
(**A**) Phylogenetic relationships of the 26 identified benzoxazinoids (BXs) biosynthetic gene cluster (BGC) gene members in maize. (**B**) The exon–intron structure of the 26 BGC gene members in maize. Orange boxes represent exons, gray lines represent introns, and bluish-violet boxes represent upstream or downstream untranslated regions (UTR). (**C**) Conserved motifs distribution of these BX proteins. (**D**) The detailed sequences and logos of the conserved motifs.

**Figure 4 ijms-25-07460-f004:**
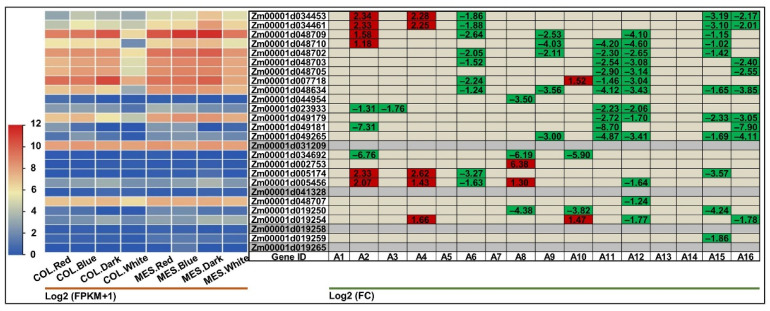
Heatmaps of expression patterns of the 26 genes in benzoxazinoids (BXs) biosynthetic gene cluster (BGC) identified in coleoptiles and mesocotyls of Zheng58 seedlings at ten days after germination in red, blue, white light, and darkness analyzed by RNA-sequencing (RNA-Seq). The left panel represents the expression patterns of different genes under different light conditions, while the numbers in right panel represent the log2 (FC) of corresponding differentially expressed genes (DEGs) compared to their controls in left panel, respectively. The bar represents the criterion of log2 (FPKM + 1). COL.Red, COL.Blue, COL.White, and COL.Dark are coleoptiles in red light, blue light, white light, and darkness, respectively; MES.Red, MES.Blue, MES.White, and MES.Dark are mesocotyls in red light, blue light, white light, and darkness, respectively. A1~A16 are MES.Red_vs_ MES.Blue, MES.Red_MES.Dark, MES.Red_vs_MES.White, MES.Blue_vs_MES.Dark, MES.Blue_vs_ MES.White, MES.Dark_vs_MES.White, COL.Red_vs_COL.Blue, COL.Red_vs_COL.Dark, MES.Red_vs_MES.White, COL.Blue_vs_COL.Dark, COL.Blue_vs_COL.White, MES.Red_vs_ COL.Red, MES.Blue_vs_COL.Blue, MES.Dark_vs_COL.Dark, and MES.White_vs_COL.White comparisons, respectively. FPKM is the fragments per kilobase of transcript per million mapped read; FC is the fold change.

**Figure 5 ijms-25-07460-f005:**
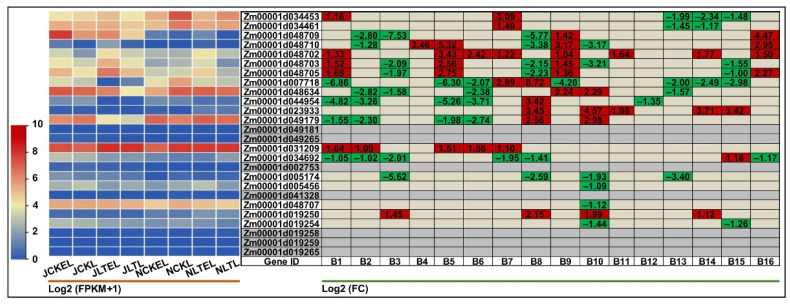
Heatmaps of expression patterns of the 26 genes in benzoxazinoids (BXs) biosynthetic gene cluster (BGC) identified in young leaves of N192 and Ji853 seedlings with or without 2 μM 24-epibrassinolide (EBR) at 25 and 15 °C conditions analyzed by RNA-sequencing (RNA-Seq). The left panel represents the expression patterns of different genes under different treatments, while the numbers in right panel represent the log2 (FC) of corresponding differentially expressed genes (DEGs) compared to their controls in left panel, respectively. The bar represents the criterion of log2 (FPKM + 1). JCKL, JCKEL, JLTL, and JLTEL are young leaves of Ji853 with 0 μM EBR at 25 °C, with 2 μM EBR at 25 °C, with 0 μM EBR at 15 °C, and with 2 μM EBR at 15 °C, respectively; NCKL, NCKEL, NLTL, and NLTEL are young leaves of N192 with 0 μM EBR at 25 °C, with 2 μM EBR at 25 °C, with 0 μM EBR at 15 °C, and with 2 μM EBR at 15 °C, respectively. B1~B16 are JCKEL_vs_JLTEL, JCKEL_vs_JLTL, JCKEL_vs_NCKEL, JCKL_vs_JCKEL, JCKL_vs_JLTEL, JCKL_vs_JLTL, JCKL_vs_ NCKL, JLTEL_vs_NLTEL, JLTL_vs_JLTEL, JLTL_vs_NLTL, NCKEL_vs_NLTEL, NCKEL_vs_NLTL, NCKL_vs_NCKEL, NCKL_vs_NLTEL, NCKL_vs_NLTL, and NLTL_vs_NLTEL comparisons, respectively. FPKM is the fragments per kilobase of transcript per million mapped read; FC is the fold change.

**Figure 6 ijms-25-07460-f006:**
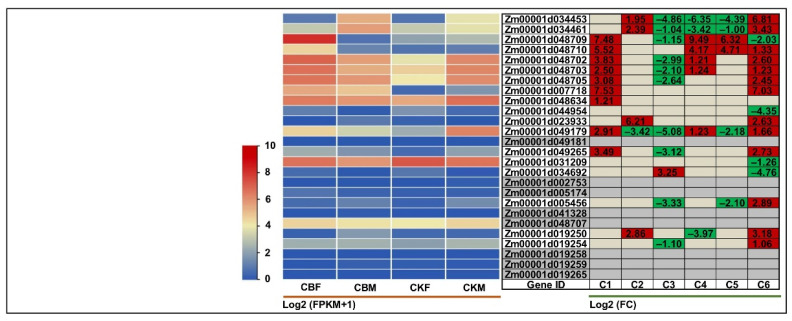
Heatmaps of expression patterns of the 26 genes in benzoxazinoids (BXs) biosynthetic gene cluster (BGC) identified in whorl leaves of F1227 and M0800 plants to feeding and non-feeding conditions by Asian corn borer (ACB) analyzed by RNA-sequencing (RNA-Seq). The left panel represents the expression patterns of different genes under different treatments, while the numbers in right panel represent the log2 (FC) of corresponding differentially expressed genes (DEGs) compared to their controls in left panel, respectively. The bar represents the criterion of log2 (FPKM + 1). CKF and CBF are F1227 whorl leaves to non-feeding and feeding by ACB, respectively; CKM and CBM are M0800 whorl leaves to non-feeding and feeding by ACB, respectively. C1~C6 are CKF_vs_CBF, CKM_vs_CBM, CKM_vs_CKF, CBM_vs_CBF, CKM_vs_CBF, and CKF_vs_CBM comparisons, respectively. FPKM is the fragments per kilobase of transcript per million mapped read; FC is the fold change.

**Figure 7 ijms-25-07460-f007:**
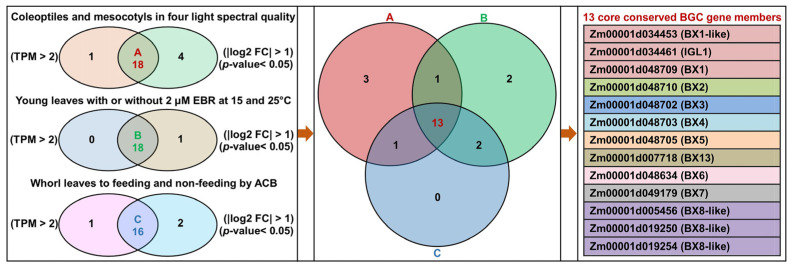
Identification of 13 core conserved differentially expressed genes (DEGs) in BGC among three independent RNA-Seq datasets. (A, red font) the common BGC gene members that show both transcripts per million (TPM) value > 2 in single sample and |log2 FC (fold-change)| > 1 and *p*-value < 0.05 in single comparison via the RNA-Seq for the coleoptiles and mesocotyls of Zheng58 seedlings at ten days after germination in red, blue, white light, and darkness. (B, green font) the common BGC gene members that show both TPM value > 2 in single sample and |log2 FC| > 1 and *p*-value < 0.05 in single comparison via the RNA-Seq for the young leaves of Ji853 and N192 seedlings at V3 stage with or without 2 μM 24-epibrassinolide (EBR) at 25 and 15 °C. (C, blue font) the common BGC gene members that show both TPM value > 2 in single sample and |log2 FC| > 1 and *p*-value < 0.05 in single comparison via the RNA-Seq for the whorl leaves of F1227 and M0800 plants at VT stage to feeding and non-feeding by Asian corn borer (ACB). FC is the fold change.

**Figure 8 ijms-25-07460-f008:**
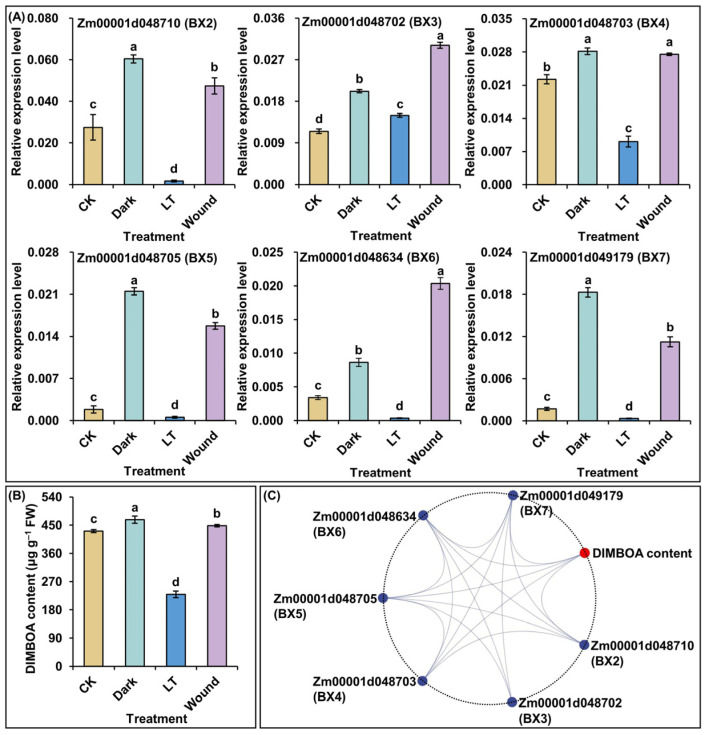
(**A**) The relative expression levels of six core conserved benzoxazinoids (BXs) biosynthetic gene cluster (BGC) gene members in young leaves of 14-day-old F1227 seedlings under four treatments, including control (CK; seedlings growing in normal environment for 14 days), darkness (dark; etiolated seedlings growing in darkness for 14 days), low-temperature (LT; 7-day-old seedlings growing at 10 °C for 7 days), and wound (wound; three leaf wounds of 3 × 3 mm^2^ were made with a scalpel on 13-day-old seedling for 24 h). Different lowercase letters represent significant differences (*p* < 0.05) by analysis of variance (ANOVA). (**B**) The DIMBOA content in young leaves of 14-day-old F1227 seedlings under corresponding four treatments. Different lowercase letters represent significant differences (*p* < 0.05) by ANOVA. (**C**) Interactive ring Pearson correlation diagram between the relative expression levels of six core BGC gene members and DIMBOA content in young leaves of 14-day-old F1227 seedlings under corresponding four treatments. Lines represent significantly positive Pearson correlation (*p* < 0.05).

## Data Availability

Data are contained within the article and Appendix A.

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
