# Peer review of "Benzoxazinoids Biosynthetic Gene Cluster Identification and Expression Analysis in Maize under Biotic and Abiotic Stresses"

_ijms, 2024, doi:10.3390/ijms25137460_

Round 1
Reviewer 1 Report
Comments and Suggestions for Authors
-Line 24, do not start with a number. Write it in letters.
-Line 26, check the english (verified!).
-Line 59, the patronymic should be added to each plant species mentioned in the text.
-Why are not BXs defined in the text as alkaloids? They are characterized by the typical N atom into its structure.
-the evolutionary aspect of the accumulation of these compounds in the plant should be better explained, together with the fact that domesticated varieties maybe cointain low levels of these substances. Is it known?
-line 509, check the full stop at the beginning of the sentence.
-in discussion the authors should better comment their data, that is the expression of BXs in presence of environmental stressors. In particular, it should be mentioned that usually other class of compounds are accumulated (flavonoids, phenols), while in maize alkaloids play a key role (link this with the 4th comment reported above). See for instance other species from other families and make some comparisons: Plant Phenolics in Biotic Stress Management. Singapore: Springer Nature Singapore, 2024. p. 121-148; Antioxidants, 2021, 10.07: 1048; Journal of plant nutrition, 2017, 40.18: 2619-2630; Journal of Medicinal Plants, 2019, 7.3: 99-106.
-In the core motifs, A letter should be coloured differently (e.g., red), because C and A are both blue and it is difficult to see the differences.
Comments on the Quality of English LanguageMinor mistakes. Please, revise the text.
Author Response
Dear Editor and Reviewers
Thank you for your letter of – and for the referee’s comments concerning our manuscript, “Benzoxazinoids Biosynthetic Gene Cluster Identification and Expression Analysis in Maize under Biotic and Abiotic Stresses (Manuscript ID: ijms-3066859)”. We have carefully studied these comments and have made corresponding corrections to the manuscript, which we describe in detail below. We would like to re-submit the manuscript and that for possible publication on the Special Issue: “Recent Advances in Maize Stress Biology” of International Journal of Molecular Sciences. Thank you very much for your time and consideration.
Editor:
Your manuscript has now been reviewed by experts in the field. Please find your manuscript with the referee reports at this link: https://susy.mdpi.com/user/manuscripts/resubmit/ca120c47b45da924f 3e41bd71cfd2f7d. Please revise the manuscript according to the referees' comments and upload the revised file within 10 days.
Thanks for the positive comments of you and all reviewers for our manuscript. As suggested, we have carefully revised and improved our manuscript using the “Track Changes” function of the manuscript at the above link. We then have re-submitted the manuscript within the allotted time.
Thank you for your consideration.
Please use the version of your manuscript found at the above link for your revisions.
Thanks for the positive comments for our manuscript. As suggested, we have carefully revised and improved our manuscript using the versions of our manuscript at the above link. We then have re-submitted the manuscript.
Thank you for your consideration.
(I) Please check that all references are relevant to the contents of the manuscript.
Thanks for your positive comments for our manuscript. As suggested, we have carefully checked all references to make sure they are relevant to the contents of the manuscript. We then have re-submitted the manuscript.
Thank you for your consideration.
(II) Any revisions to the manuscript should be highlighted, such that any changes can be easily reviewed by editors and reviewers.
Thanks for your positive comments for our manuscript. As suggested, we have carefully revised and improved our manuscript using the “Track Changes” function of the manuscript at the above link. We then have re-submitted the manuscript within the allotted time.
Thank you for your consideration.
(III) Please provide a cover letter to explain, point by point, the details of the revisions to the manuscript and your responses to the referees’ comments.
Thanks for your positive comments for our manuscript. As suggested, we have carefully revised and improved our manuscript. In addition, we have prepared a detailed response letter to all reviewers for each point, and then have re-submitted the manuscript.
Thank you for your consideration.
(IV) If the reviewer(s) recommended references, please critically analyze them to ensure that their inclusion would enhance your manuscript. If you believe these references are unnecessary, you should not include them.
Thanks for your positive comments for our manuscript. As suggested, we have carefully checked and revised the References. At the same time, we also have re-added two new references to enhance the quality of our manuscript. We then have re-submitted the manuscript.
Thank you for your consideration.
(V) If you found it impossible to address certain comments in the review reports, please include an explanation in your appeal.
Thanks for your positive comments for our manuscript. As suggested, we have carefully revised and improved our manuscript. In addition, we have prepared a detailed response letter to all reviewers for each point, and then have re-submitted the manuscript.
Thank you for your consideration.
(VI) The revised version will be sent to the editors and reviewers.
Thanks for your positive comments for our manuscript. As suggested, we have submitted the revised version to the editorial office as requested, which will be sent to the editors and reviewers.
Thank you for your consideration.
If one of the referees has suggested that your manuscript should undergo extensive English revisions, please address this issue during revision. We propose that you use one of the editing services listed at https://www.mdpi.com/authors/english or have your manuscript checked by a colleague fluent in English writing.
Thanks for the positive comments of you and all reviewers for our manuscript. As suggested, and under the guidance of Professor Xiquan Gao (who is a famous expert in maize, his English is very idiomatic; https://mp.weixin.qq.com/s/6Hbtdq50MbR4vh4qh6fBFg), we have further revised and improved the English language of the manuscript. We then re-submitted the manuscript.
Thank you for your consideration.
We would like to draw your attention to the status of this invitation “Publish Author Biography on the webpage of the paper”-https://susy.mdpi.com/user/manuscript/author_biography/ca120c47b45da 924f3e41bd71cfd2f7d. If you decide to publish your biography, please remember to fill in it before your paper is accepted.
Thanks for your invitation, we decided not to publish our biography.
Thank you for your consideration.
Please do not hesitate to contact us if you have any questions regarding the revision of your manuscript or if you need more time. We look forward to hearing from you soon.
Thanks for your positive comments for our manuscript. As suggested, we have carefully revised and improved the manuscript using the “Track Changes” function of our manuscript at the above link. We then have re-submitted the manuscript within the allotted time.
Thank you for your consideration.
Reviewer 1:
-Line 24, do not start with a number. Write it in letters.
Thanks for your positive comments. As suggested, we have revised the content, namely “Thirteen differentially expressed genes (DEGs) with high and specific expression levels were commonly detected among three RNA-Seq, as core conserved BGC members for regulating BXs biosynthesis under multiple abiotic/biotic stimulates.” We then have re-submitted the manuscript.
Thank you for your consideration.
-Line 26, check the english (verified!).
Thanks for your positive comments. As suggested, the “varified” was revised to the “verified” in the Abstract section of the manuscript. We then have re-submitted the manuscript.
Thank you for your consideration.
-Line 59, the patronymic should be added to each plant species mentioned in the text.
Thanks for your positive comments. That's a very good question. As suggested, we have provided the corresponding information, namely: “Sesamia nonagrwides (Lepidoptera: Noctuidae)”. In addition, we also have provided the corresponding information: Asian corn borer (ACB; Ostrinia furnacalis; Lepidoptera, Pyralidae), and hairy vetch (Vicia villosa Roth) in our manuscript. We then have re-submitted the manuscript.
Thank you for your consideration.
-Why are not BXs defined in the text as alkaloids? They are characterized by the typical N atom into its structure.
Thanks for your positive comments. That's a very good question. Yes, I completely agree with you on the definition of benzoxazinoids (BXs), i.e., BXs could be defined as alkaloids. At the same time, in this manuscript, the biosynthetic pathway of BXs was drew in detail from Figure 1, we found that BXs were a group of cyclic hydroxamic acids with the 1,4-benzoxazin-3-one skeleton (Figure 1), they thus commonly known as a class of indole-derived plant chemical defense compounds. Interestingly, many previous reports also supports our viewpoint. Such as, Stahl reported that BXs were abundant indole-derived specialized metabolites in several monocot crop species (Stahl, E. New insights into the transcriptional regulation of benzoxazinoid biosynthesis in wheat. J. Exp. Bot. 2022, 73, 5358–5360.). Wu et al. and Zhou et al. reported that BXs were a class of indole-derived plant metabolites that function in defense against numerous pests and pathogens (Wu, D.Y.; Jiang, B.W.; Ye, C.Y.; Timko, M.P.; Fan, L.J. Horizontal transfer and evolution of the biosynthetic gene cluster for benzoxazinoids in plants. Plant Comm. 2022, 3, 100320. Zhou, S.Q.; Richter, A.; Jander, G. Beyond defense: Multiple functions of benzoxazinoids in maize metabolism. Plant & Cell Physiol. 2018, 59, 1528–1537.). Mwendwa et al. also reported that BXs were a class of indole-derived plant defence chemicals containing compounds with a 2-hydroxy-2H-1,4-benzoxazin-3(4H)-one skeleton (Mwendwa, J.M.; Weston, P.A.; Weidenhamer, J.D.; Fomsgaard, I.S.; Wu, H.W.; Gurusinghe, S.; Weston, L.A. Metabolic profiling of benzoxazinoids in the roots and rhizosphere of commercial winter wheat genotypes. Plant Soil 2021, 466, 467–489.).
Thank you for your consideration.
Figure 1. The skeleton structure of benzoxazinoids (BXs), major BXs and corresponding enzymes, including IGL1, BX1, BX2, BX3, BX4, BX5, BX6, BX7, BX8, BX9, and BX13, catalyzing their biosynthesis in maize.
-the evolutionary aspect of the accumulation of these compounds in the plant should be better explained, together with the fact that domesticated varieties maybe cointain low levels of these substances. Is it known?
Thanks for your positive comments. That's a very good question. In a previous study, the complete benzoxazinoids (BXs) clusters (containing genes BX1–BX5 and BX8) in three genera (Zea, Echinochloa, and Dichanthelium) of Panicoideae and partial clusters in Triticeae were identified, and showing that the ancient BX cluster was putatively gained by the Triticeae ancestor via horizontal transfer (HT) from the ancestral Panicoideae and later separated into multiple segments on different chromosomes. BX6 appears to have been under less constrained selection compared with the BX cluster during the evolution of Panicoideae. Further investigations indicated that purifying selection and polyploidization have shaped the evolutionary trajectory of BX clusters in the grass family (Wu, D.Y.; Jiang, B.W.; Ye, C.Y.; Timko, M.P.; Fan, L.J. Horizontal transfer and evolution of the biosynthetic gene cluster for benzoxazinoids in plants. Plant Comm. 2022, 3, 100320.).
Another early study also showed that "Evolution Canyon" (ECI), Mount Carmel, Israel, is an optimal natural microscale model for unraveling evolution-in-action, highlighting the evolutionary processes of biodiversity evolution, adaptation, and incipient sympatric speciation. A major model organism in ECI is the tetraploid wild emmer wheat, Triticum dicoccoides (TD), the progenitor of cultivated emmer and durum wheat. TD displays dramatic interslope adaptive evolutionary divergence on the tropical, savannoid-hot and dry south-facing, "African" slope (AS), and on the temperate, forested, cool and humid, north-facing, "European" slope (ES), separated on average by 250 m. Further analysis found that higher concentrations of 2,4-dihydroxy-1,4-benzoxazin-3-one (DIBOA) and 2,4-dihydroxy-7-methoxy-1,4-benzoxazin-3- one (DIMBOA) were found in seedlings growing in the same greenhouse from seeds collected from the cool and humid forested ES, whereas the seedlings of seeds collected from the savannoid AS (both in root and shoot tissues), showed no DIMBOA. Remarkably, only DIBOA appears in both shoots and roots of the AS seedlings. It rises to a peak and then decreases in both organs and in seedlings from both slopes. The DIMBOA, which appears only in the ES seedlings, rises to a peak and decreases in the shoot, but increased and remained in a plateau in the root (Yuval, B.A.; Avigdor, B.; Dvir, F.; Eviatar, N. Adaptive evolution of benzoxazinoids in wild emmer wheat, Triticum dicoccoides, at “Evolution Canyon”, Mount Carmel, Israel. PLoS ONE 2018, 13, e0190424.).
In addition, to reveal BXs accumulation in maize evolution, previous our team measured the DIMBOA contents among 310 elite maize inbred lines from different ecological environments in Gansu Province, China. The results showed that the average DIMBOA content of 310 inbred lines seedlings in Zhangye (E1) environment ranged from 8.66 (T58) to 528.88 μg g−1 FW (ZY19-Jiu1101) with an overall mean of 94.21 μg g−1 FW; similarly, the average DIMBOA content of these seedlings in Longxi (E2) environment varied from 8.84 (T58) to 493.40 μg g−1 FW (RX20-1006) with an overall mean of 93.26 μg g−1 FW. Meanwhile, the genetic variation coefficient (CVg) of DIMBOA content among 310 maize inbred lines in both E1 and E2 environments were 108.47 and 106.22%, respectively (Niu, Y.N.; Zhao, X.Q.; Chao, W.; Lu, P.N.; Bai, X.D.; Mao, T.T. Genetic variation, DIMBOA accumulation, and candidate gene identification in maize multiple insect-resistance. Int. J. Mol. Sci. 2023, 24, 2138.).
Based on the above considerations, breeders have fully considered the breeding application of BXs in recent years, and the “Dongdan1775” and “Dongdan1331” varieties of maize, with higher BXs contents have been successfully developed in 2018 and 2016, China.
Thank you for your consideration.
-line 509, check the full stop at the beginning of the sentence.
Thanks for your positive comments. As suggested, we have checked and deleted the full stop, and the revised content was that “The 20 soaked seeds of Ji853 (chilling-sensitive) [56] and N192 (chilling-tolerant) with ddH2O, were sown in plastic boxes, and cultured in a growth chamber (25±0.5°C, 300 μM m-2 s-1 white light, 12 h photoperiod).” We then have re-submitted the manuscript.
Thank you for your consideration.
-in discussion the authors should better comment their data, that is the expression of BXs in presence of environmental stressors. In particular, it should be mentioned that usually other class of compounds are accumulated (flavonoids, phenols), while in maize alkaloids play a key role (link this with the 4th comment reported above). See for instance other species from other families and make some comparisons: Plant Phenolics in Biotic Stress Management. Singapore: Springer Nature Singapore, 2024. p. 121-148; Antioxidants, 2021, 10.07: 1048; Journal of plant nutrition, 2017, 40.18: 2619-2630; Journal of Medicinal Plants, 2019, 7.3: 99-106.
Thanks for your positive comments. That's a very good question. Just as you think, the expression of benzoxazinoids (BXs) in presence of environmental stressors, including low temperature, light spectral quality, 24-epibrassinolide (EBR) induction, and Asian corn borer (ACB) infestation were better comment in the Results and Discussion sections of our manuscript.
In particular, it should be mentioned that usually other class of compounds are accumulated (flavonoids, phenols), while alkaloids including BXs play a key role in different crops under various biotic and abiotic stresses.
Duan et al. reported that in wheat most of detected 102 metabolites belonged to the flavonoid and phenolic acid, which were significantly changed because of Fusarium crown rot (FCR) infection. Meanwhile based on combined data from gene expression and metabolite profiles, two metabolites, benzoxazolin-2-one (BOA) and 6-methoxy-benzoxazolin-2-one (MBOA), involved in the benzoxazinoid-biosynthesis pathway, were tested for their effects on FCR resistance. Both BOA and MBOA significantly reduced fungal growth in vitro and in vivo, and, thus, a higher content of BOA and MBOA in wheat 04z36 genotypes may contribute to FCR resistance (Duan, S.N.; Jin, J.J.; Gao, Y.T.; Jin, C.L.; Mu, J.Y.; Zhen, W.C.; Sun, Q.X.; Xie, C.J.; Ma, J. Integrated transcriptome and metabolite profiling highlights the role of benzoxazinoids in wheat resistance against Fusarium crown rot. Crop J. 2022, 10, 407–417.). Walker et al. also reported that maize seeds were inoculated singly with selected strains from bacterial genera Pseudomonas and Azospirillum or mycorrhizal genus Glomus, or with these strains combined two by two or all three together. At 16 days, maize root methanolic extracts were analyzed by RP-HPLC and secondary metabolites (phenolics, flavonoids, xanthones, benzoxazionoids, etc.) identified by LC/MS. Results Inoculation did not impact on plant biomass but resulted in enhanced total root surface, total root volume and/or root number in certain inoculated treatments, at reduced fertilization. Inoculation led to qualitative and quantitative modifications of root secondary metabolites, particularly benzoxazinoids and diethylphthalate. These modifications depended on fertilization level and microorganism(s) inoculated (Walker, V.; Couillerot, O.; Felten, A.V.; Bellvert, F.; Jansa, J.; Maurhofer, M.; Bally, R.; Moenne-Loccoz, Y.; Comte, G. Variation of secondary metabolite levels in maize seedling roots induced by inoculation with Azospirillum, Pseudomonas and Glomus consortium under field conditions. Plant Soil 2012, 356–163.).
Similarly, in present study, the Ji853 (chilling-sensitive) and N192 (chilling-tolerant) seedling at V3 stage were cultured for seven days with or without 2 μM exogenous EBR at 15 and 25°C environments. Next the young leaves of both maize seedlings under four treatments, i.e., JCKL (leaves of Ji853 with 0 μM EBR at 25°C), JCKEL (leaves of Ji853 with 2 μM EBR at 25°C), JLTL (leaves of Ji853 with 0 μM EBR at 15°C), JLTEL (leaves of Ji853 with 2 μM EBR at 15°C), NCKL (leaves of N192 with 0 μM EBR at 25°C), NCKEL (leaves of N192 with 2 μM EBR at 25°C), NLTL (leaves of N192 with 0 μM EBR at 15°C), NLTEL (leaves of N192 with 2 μM EBR at 15°C), with three biological replicates, 24 sample in total were used for RNA-Seq. The results showed that the differentially expressed genes (DEGs) were significantly enriched in multiple KEGG pathways, such as Flavonoid biosynthesis and Benzoxazinoid biosynthesis in JCKL_vs_JCKEL, JCKL_vs_JLTL, JCKL_vs_JLTEL, and JLTL_vs_JLTEL comparisons (Fig. S1, This result was not shown in the manuscript).
Fig. S1. The KEGG enrichment annotation of differentially expressed genes (DEGs) among four groups in Ji853 seedlings.
Fig. S2. The KEGG enrichment annotation of differentially expressed genes (DEGs) among two groups in F117 and M0800 plants.
In addition, in current study, the whorl leaves of F1227 (moderate insect-resistant) and M0800 (insect-susceptible) plants to feeding and non-feeding conditions by ACB at VT stage for 24 h. Next, the whorl leaves of both maize plants under two treatments, i.e., CKF (whorlleaves of F1227 to non-feeding by ACB), CBF (whorl leaves of F1227 to feeding by ACB), CKM (whorl leaves of M0800 to non-feeding by ACB), CBM (whorl leaves of M0800 to feeding by ACB), with three biological replicates, 12 sample in total were used for RNA-Seq. The results showed that the DEGs were significantly enriched in multiple KEGG pathways, such as Flavonoid biosynthesis and Benzoxazinoid biosynthesis in CKF_vs_CBF group (Fig. S2, This result was not shown in the manuscript).
Considering above interesting ecological functions of BXs, revealing the molecular mechanisms of BXs biosynthesis and decomposition by different stimulations will be beneficial to elucidate their evolutionary adaptability and breeding programs in maize. Fortunately, the genetic variation in maize BXs accumulation is greatly facilitated by recent genomics advances, facilitating the discovery of previously unknown genes and functions of BXs biosynthesis. In the context of these considerations, present study aims to (i) identify BXs biosynthetic gene cluster (BGC) gene members in maize, and comprehensively analyze their phylogenetic relationships, sequence features, chromosome distributions, and subcellular localization; (ii) analyze the expression patterns of all BGC gene members in various biotic/abiotic-induced tissues of maize via RNA-sequencing (RNA-Seq); (iii) verify the relative expression patterns of the core conserved BGC genes and DIMBOA accumulation in response to different biotic/abiotic stresses by quantitative real-time PCR (qRT-PCR) and HPLC. The findings may provide valuable information for further functional characterization of BGC genes and their breeding application in maize. Therefore, the relationships between BXs accumulation and other compounds (flavonoids, phenols) accumulation were ignored in this manuscript.
Thank you for your consideration.
-In the core motifs, A letter should be coloured differently (e.g., red), because C and A are both blue and it is difficult to see the differences.
Thanks for your positive comments. As suggested, in the ten core motifs, the color of C letter (blue) have been replaced by the red color. Then we have re-provided the new Figure 3D and re-submitted the manuscript.
Thank you for your consideration.
Figure 3. (D) The detailed sequences and logos of the conserved motifs.
Comments on the Quality of English Language: Minor mistakes. Please, revise the text.
Thanks for your positive comments. As suggested, we have carefully revised and improved the English language quality of the manuscript using the “Track Changes” function. We then have re-submitted the manuscript.
Thank you for your consideration.
Open Review: I would not like to sign my review report.
Thanks for your positive comments.
Thank you for your consideration.
Quality of English Language: Minor editing of English language required.
Thanks for your positive comments. As suggested, we have carefully revised and improved the English language quality of the manuscript using the “Track Changes” function. We then have re-submitted the manuscript.
Thank you for your consideration.
Does the introduction provide sufficient background and include all relevant references? Can be improved.
Thanks for your positive comments. As suggested, we have carefully revised and improved the Introduction section of the manuscript using the “Track Changes” function. We then have re-submitted the manuscript.
Thank you for your consideration.
Is the research design appropriate? Yes.
Thanks for your positive comments.
Thank you for your consideration.
Are the methods adequately described? Yes.
Thanks for your positive comments.
Thank you for your consideration.
Are the results clearly presented? Yes.
Thanks for your positive comments.
Thank you for your consideration.
Are the conclusions supported by the results? Can be improved.
Thanks for your positive comments. As suggested, we have carefully revised and improved the Conclusion section of the manuscript using the “Track Changes” function. We then have re-submitted the manuscript.
Thank you for your consideration.
Reviewer 2:
> Title of this manuscript is too lengthy, need to make it brief and understanding.
Thanks for your positive comments. As suggested, we have revised the Title of the manuscript, namely: “Benzoxazinoids Biosynthetic Gene Cluster Identification and Expression Analysis in Maize under Biotic and Abiotic Stresses”, to make it brief and understand. We then have re-submitted the manuscript.
Thank you for your consideration.
> Abstract written in good format and style with the Rythm but authors need to clearer the conclusion and future recommendations for reviewers.
Thanks for your positive comments. That's a very good question. As suggested, we have carefully revised and improved the Abstract section of the manuscript, namely: “Benzoxazinoids (BXs) are unique bioactive metabolites with protective and allelopathic properties in maize in response to diverse stresses. The production of BXs involves the fine regulations of BXs biosynthetic gene cluster (BGC). However, little is known about whether and how the expression pattern of BGC members is impacted by biotic and abiotic stresses. Here, maize BGC was systemically investigated and 26 BGC gene members were identified on seven chromosomes, for which Bin 4.00-4.01/4.03-4.04/7.02 were the most enriched regions. All BXs proteins were clearly divided into three classes and seven subclasses, and ten conserved motifs were further identified among these proteins. These proteins were localized in the subcellular compartments of chloroplast, endoplasmic reticulum, or cytoplasmic, where their catalytic activities were specifically executed. Three independent RNA-sequencing (RNA-Seq) analyses revealed that the expression profiles of the majority of BGC gene members were distinctly affected by multiple treatments, including light spectral quality, low-temperature, 24-epibrassinolide induction, and Asian corn borer infestation. Thirteen differentially expressed genes (DEGs) with high and specific expression levels were commonly detected among three RNA-Seq, as core conserved BGC members for regulating BXs biosynthesis under multiple abiotic/biotic stimulates. Moreover, the quantitative real-time PCR (qRT-PCR) verified that six core conserved genes in BGC were significantly differentially expressed in leaves of seedlings upon four treatments, which caused significant increases of 2,4-dihydroxy-7-methoxy-1,4-benzoxazin-3-one (DIMBOA) content under darkness and wound treatments, whereas a clear decrease of DIMBOA content was observed under low-temperature treatment. In conclusion, the changes of BXs metabolites in maize were regulated by BGC gene members in multiple stress presences. Therefore, identification of key genes associated with BXs accumulation under biotic/abiotic stresses will provide valuable gene resources for breeding maize varieties with enhanced capability to adapt environment stresses.”
In addition, we have also provided the clearer conclusion and future recommendations for reviewers, namely: “In conclusion, the changes of BXs metabolites in maize were regulated by BGC gene members in multiple stress presences. Therefore, identification of key genes associated with BXs accumulation under biotic/abiotic stresses will provide valuable gene resources for breeding maize varieties with enhanced capability to adapt environment stresses.”
We then have re-submitted the manuscript.
Thank you for your consideration.
> Keyword should not belong to title or abstract part. Need to rephrase them.
Thanks for your positive comments. As suggested, we have carefully revised and improved the Keywords of the manuscript, namely: “biotic/abiotic stresses; benzoxazinoids biosynthetic gene cluster; maize; gene expression; RNA-sequencing; phylogenetic analysis; gene structure”. We then have re-submitted the manuscript.
Thank you for your consideration.
> In Figure 2b red dotted line showing the same trend as the top of each bar, I preferred to change the bar length to for frequency. So, reader can easily judge the graphs.
Thanks for your positive comments. That's a very good question. Just as you think, in Figure 2B, the red dotted line represented the “Frequency of BGC gene member (%)”.
Thank you for your consideration.
Figure 2B. The summary of 26 benzoxazinoids (BXs) biosynthetic gene cluster (BGC) gene members on ten chromosomes.
>In Figure 3A Its better to remove the UTR from the Gene structure to make it more align to each gene.
Thanks for your positive comments. That's a very good question. It is well known that the sequence of gene included CDS, UTR, and Intron. Generally, the gene structure analysis of gene family included CDS, UTR, and Intron. For instance, Zhu et al. analyzed the gene structure of LOX gene family in potato (Zhu, J.Y.; Chen, L.M.; Li, Z.T.; Wang, W.L.; Qi, Z.Y.; Li, Y.M.; Liu, Y.H.; Liu, Z. Genome-wide identification of LOX gene family and its expression analysis under abiotic stress in potato (Solanum tuberosum L.). Int. J. Mol. SCi. 2024, 25, 3487.). Therefore, we would like to keep the UTR sequence in gene structure analysis of maize benzoxazinoids (BXs) biosynthetic gene cluster (BGC) gene members (Figure 3B).
Thank you for your consideration.
Figure 3. (A) Phylogenetic relationships of the 26 identified benzoxazinoids (BXs) biosynthetic gene cluster (BGC) gene members in maize. (B) The exon-intron structure of the 26 BGC gene members in maize. Orange boxes represented exons, gray lines represented introns, and bluish violet boxes represented upstream or downstream untranslated regions (UTR). (C) Conserved motifs distribution of these BXs proteins.
> In Fig 3B another way if authors arranged the Motif analysis with the Gene Structure will be more understandable for reviewers.
Thanks for your positive comments. That's a very good question. As suggested, we have re-drawn the Figure 3A, 3B, and 3C. We then have re-submitted the manuscript.
Thank you for your consideration.
Figure 3. (A) Phylogenetic relationships of the 26 identified benzoxazinoids (BXs) biosynthetic gene cluster (BGC) gene members in maize. (B) The exon-intron structure of the 26 BGC gene members in maize. Orange boxes represented exons, gray lines represented introns, and bluish violet boxes represented upstream or downstream untranslated regions (UTR). (C) Conserved motifs distribution of these BXs proteins.
> Figure 4, 5 and 6 the bars for log2 can't see here, as in general I can't what that value belongs to and which part it linked with.
Thanks for your positive comments. That's a very good question. In our manuscript, the bars in Figure 4, 5, and 6 represents the Log2 (FPKM+1) of 26 corresponding benzoxazinoids (BXs) biosynthetic gene cluster (BGC) gene members under different biotic/abiotic stresses. Therefore, the corresponding description has been added to figure legend, namely: “Figure 4. Heatmaps of expression patterns of the 26 genes in benzoxazinoids (BXs) biosynthetic gene cluster (BGC) identified in coleoptiles and mesocotyls of Zheng58 seedlings at ten days after germination in red, blue, white light, and darkness analyzed by RNA-sequencing (RNA-Seq). The left panel represents the expression patterns of different genes under different light conditions, while the numbers in right panel represent the log2 (FC) of corresponding differentially expressed genes (DEGs) compared to their controls in left panel, respectively. The bar represents the criterion of log2 (FPKM+1).” “Figure 5. Heatmaps of expression patterns of the 26 genes in benzoxazinoids (BXs) biosynthetic gene cluster (BGC) identified in young leaves of N192 and Ji853 seedlings with or without 2 μM 24-epibrassinolide (EBR) at 25 and 15 °C conditions analyzed by RNA-sequencing (RNA-Seq). The left panel represents the expression patterns of different genes under different treatments, while the numbers in right panel represent the log2 (FC) of corresponding differentially expressed genes (DEGs) compared to their controls in left panel, respectively. The bar represents the criterion of log2 (FPKM+1).” “Figure 6. Heatmaps of expression patterns of the 26 genes in benzoxazinoids (BXs) biosynthetic gene cluster (BGC) identified in whorl leaves of F1227 and M0800 plants to feeding and non-feeding conditions by Asian corn borer (ACB) analyzed by RNA-sequencing (RNA-Seq). The left panel represents the expression patterns of different genes under different treatments, while the numbers in right panel represent the log2 ((FC) of corresponding differentially expressed genes (DEGs) compared to their controls in left panel, respectively. The bar represents the criterion of log2 (FPKM+1).” We then have re-submitted the manuscript.
Similarly, heatmaps of differentially expressed genes (DEGs) encoding SOD, POD, CAT, GPX, APX, and GST was drew in previous study (Liu, J.P.; Tang, X.; Zhang, H.H.; Wei, M.; Zhang, N.; Si, H.J. Transcriptome analysis of potato leaves under oxidative stress. Int. J. Mol. Sci. 2024, 5994.).
Thank you for your consideration.
> In Figure 8 my suggestion is to move the DIMBOA graphs on Side and move the relative expression graphs in two rows with 3 columns.
Thanks for your positive comments. That's a very good question. As suggested, we have re-drawn the Figure 8, as well as we have re-added the interactive ring Pearson correlation diagram between the relative expression levels of six core BGC gene members and DIMBOA content in young leaves of 14-day old F1227 seedlings under corresponding four treatments. We then have re-submitted the manuscript.
Thank you for your consideration.
Figure 8. (A) The relative expression levels of six core conserved benzoxazinoids (BXs) biosynthetic gene cluster (BGC) gene members in young leaves of 14-day old F1227 seedlings under four treatments, including control (CK; seedlings growing in normal environment for 14 days), darkness (Dark; etiolated seedlings growing in darkness for 14 days), low-temperature (LT; seven-day old seedlings growing at 10°C for seven days), and wound (Wound; three leaf wounds of 3×3 mm were made with a scalpel on 13-day old seedling for 24 h). Different lowercase letters represent significant differences (p < 0.05) by analysis of variance (ANOVA). (B) The DIMBOA content in young leaves of 14-day old F1227 seedlings under corresponding four treatments. Different lowercase letters represent significant differences (p < 0.05) by ANOVA. (C) Interactive ring Pearson correlation diagram between the relative expression levels of six core BGC gene members and DIMBOA content in young leaves of 14-day old F1227 seedlings under corresponding four treatments. Lines represent significantly positive Pearson correlation (p < 0.05).
> Metrial and method are well written with everything clear.
Thanks for your positive comments.
Thank you for your consideration.
Open Review: I would like to sign my review report.
Thanks for your positive comments.
Thank you for your consideration.
Quality of English Language: I am not qualified to assess the quality of English in this paper.
Thanks for your positive comments.
Thank you for your consideration.
Does the introduction provide sufficient background and include all relevant references? Yes.
Thanks for your positive comments.
Thank you for your consideration.
Is the research design appropriate? Yes.
Thanks for your positive comments.
Thank you for your consideration.
Are the methods adequately described? Yes.
Thanks for your positive comments.
Thank you for your consideration.
Are the results clearly presented? Yes.
Thanks for your positive comments.
Thank you for your consideration.
Are the conclusions supported by the results? Yes.
Thanks for your positive comments.
Thank you for your consideration.
Reviewer 3:
In this manuscript, the authors aim to: (i) identify BXs biosynthetic gene cluster (BGC) gene members in maize, and comprehensively analyze their phylogenetic relationships, sequence features, chromosome distributions, and subcellular localization; (ii) analyze the expression patterns of all BGC gene members in various biotic/abiotic-induced tissues of maize via RNA-sequencing (RNA-Seq); (iii) verify the relative expression patterns of the core conserved BGC genes in response to different biotic/abiotic stresses by quantitative real-time PCR (qRT-PCR) and HPLC."
Thanks for your positive comments.
Thank you for your consideration.
Overall, the manuscript has a great potential. However, methods are not clear and the authors seem to have used different tissues to achieve these aims. Without further clarification, these raise several concerns. I have some doubts:
Thanks for your positive comments. As suggested, we have carefully revised and improved the Materials and Methods sections of the manuscript using the “Track Changes” function, that made the methods descriptions of the study more clear and accurate, and the readers will be clearly repeat the experiments in the future. We then have re-submitted the manuscript within the allotted time.
Thank you for your consideration.
- Methods lack some details. For instance, "Then, the coleoptiles and mesocotyls of Zheng58 seedlings under four treatments, with three biological replicates, in total 24 samples were used for RNA-Seq [54]." - which treatments? Any details about Zheng58? Why was it used?
Thanks for your positive comments. As suggested, we have improved and described the experiment method in detail, namely: “The 20 soaked seeds of Zheng58 (a representative inbred line in China, is the female parent of Zhengdan958 hybrid [54]) with double-distilled water (ddH2O), were pre-cultured using germinating boxes in darkness at 22±0.5°C for five days. They were subsequently placed into plant chambers and were illuminated with lamps consisting of three light-emitting diode (LED) bars specifically designed to provide a custom spectrum, including white light (photosynthetic photon flux density (PFD) 17 μM m-2 s-1, 12 h photoperiod), red light (peak wavelength 660 nm, PFD 22 μM m-2 s-1, 12 h photoperiod), and blue light (peak wavelength 450 nm, PFD 13 μM m-2 s-1, 12 h photoperiod) in each chamber, respectively. The control was remained in the darkness. During culture, 20 mL Hoagland solution was watered to each box at 2-day intervals, the temperature was set to 22±0.5°C and relative humidity was set to 70% for ensuring adequate moisture. Then the coleoptiles and mesocotyls of Zheng58 seedlings under four treatments (darkness, white light, red light, and blue light), with three biological replicates, in total 24 samples were collected and used for RNA-Seq [55]. The published raw data of this RNA-Seq was deposited to NCBI BioProject database (accession number PRJNA917698).” We then have re-submitted the manuscript.
For this experiment, there were four treatments including darkness, white light, red light, and blue light. In addition, the elite Zheng58 inbred line was widely used in maize breeding in China, that derived from Reid heterotic group, with superior drought tolerance (Wang, Y.Z.; Nie, L.J.; Ma, J.; Zhou, B.; Han, X.H.; Cheng, J.L.; Lu, X.M.; Fan, Z.F.; Li, Y.L.; Cao, Y.Y. Transcriptome variations and network hubs controlling seed size and weight during maize seed development. Front. Plant Sci. 2022, 13, 828923.).Therefore, we have revised the corresponding content, namely: “Zheng58, a representative inbred line in China, is the female parent of Zhengdan958 hybrid [54]” ([54] Sun, J.Y.; Gao, J.L.; Wang, Z.G.; Hu, S.P.; Zhang, F.J.; Bao, H.Z.; Fan, Y.F. Maize canopy photosynthetic efficiency, plant growth, and yield response to tillage depth. Agronomy 2019, 9, 3.). We then have re-submitted the manuscript.
Thank you for your consideration.
For Ji853 (chilling-sensitive) and N192 (chilling-tolerant) , the authors wrote ". Overall, the young leaves of both maize seedlings under four treatments, with three biological replicates, in total 24 samples were used for RNA- Seq." Here leaves were rather used? Why? Again, which treatments?
Thanks for your positive comments. As suggested, we have improved and described the experiment method in detail, namely: “The 20 soaked seeds of Ji853 (chilling-sensitive) and N192 (chilling-tolerant) [56] with ddH2O, were sown in plastic boxes, and cultured in a growth chamber (25±0.5°C, 300 μM m-2 s-1 white light, 12 h photoperiod). At V3 stage of seedlings, they were subsequently placed into 25±0.5 and 15±0.5°C environment for 7 days, along with 5 mL of 2 μM EBR (CAS: 78821-43-9) spayed on seedling leaves each day. The control seedling leaves received same volume of ddH2O. 30 mL Hoagland nutrient solution was added to plastic boxes at 2-day intervals during seedlings growth period. Overall, the young leaves of both maize seedlings under four treatments (0 μM EBR at 25°C, 2 μM EBR at 25°C, with 0 μM EBR at 15°C, and 2 μM EBR at 15°C), with three biological replicates, in total 24 samples were collected and used for RNA-Seq. The new raw data of this RNA-Seq was deposited to NCBI BioProject database (accession number PRJNA1096968).” We then have re-submitted the manuscript.
For this experiment, the young leaves of both maize seedlings (Ji853 and N192) under four treatments (0 μM EBR at 25°C, 2 μM EBR at 25°C, with 0 μM EBR at 15°C, and 2 μM EBR at 15°C) were collected and used for RNA-Seq. Therefore, there were four treatments including 0 μM EBR at 25°C, 2 μM EBR at 25°C, with 0 μM EBR at 15°C, and 2 μM EBR at 15°C. At the same time, compared to chilling-tolerant N192, the PSII of chilling-sensitive Ji853 was heavy damaged under 8°C low temperature stress (Jia, A.; Wen, W.L.; Yang, D.G.; Zhang, Q.; Feng, D.D.; Wang, H.Y. Effects of low temperature and recovery on photosynthetic characteristics of maize. J. Maize Sci. 2014, 22, 92–97.). Ji853, female parent, was used for development the maize varieties of Jidan180 and Jidan261. Therefore, we have re-added the corresponding reference in our manuscript. We then have re-submitted the manuscript.
Thank you for your consideration.
For F1227 (insect-resistant) and M0800 (insect-susceptible), the authors used "the whorl leaves of F1227 and M0800 plants under two treatments, with three biological replicates, in total 12 samples were used for RNA- Seq."
Thanks for your positive comments. As suggested, we have improved and described the experiment method in detail, namely: “The seeds of F1227 (insect-resistant) and M0800 (insect-susceptible) [29] were planted at a density of 67,500 plants/hm2 with 0.4 m row spacing, on 22 April 2023, at the Longxi experimental stations, Gansu, China (34°58′N, 104°23′E, 2,074 m altitude; sandy loam), with a split block design. A plastic film (0.01 mm thick, 120 cm wide) was laid out by hand over the fields and covered the soil surface, other managements unified as field. Weather data at the site in all maize life stages were recorded (Figure S1). When the plants were grew to the VT stage, the six same instar larvae of ACB were placed in the whorl leaves of corresponding plants and were allowed to feed freely for 24 h. The corresponding leaves were protected with nets to avoid ACB dispersion. For the control plants, there were no ACB in the nets. Next, the whorl leaves of F1227 and M0800 plants under two treatments (non-feeding and feeding by ACB), with three biological replicates, in total 12 samples were used for RNA-Seq. The new raw data of this RNA-Seq was deposited to NCBI BioProject database (accession number PRJNA1121054). ” We then have re-submitted the manuscript.
For this experiment, there were two treatment including non-feeding and feeding by ACB. Furthermore, the two maize inbred lines (F1227 and M0800) were screened out from 310 elite maize germplasms at Zhangye and Longxi experimental stations, Gansu Province, China (Niu, Y.N.; Zhao, X.Q.; Chao, W.; Lu, P.N.; Bai, X.D.; Mao, T.T. Genetic variation, DIMBOA accumulation, and candidate gene identification in maize multiple insect-resistance. Int. J. Mol. Sci. 2023, 24, 2138.). Therefore, we have re-added the corresponding reference in our manuscript. We then have re-submitted the manuscript.
Additionally, to better understand the growth of F1227 and M0800 at Longxi experimental station in 2023, we have also re-added the weather data at Longxi experiment station, 2023 in the manyscript (Figure S1).
Thank you for your consideration.
Figure S1. Weather data in all maize life stages at the Longxi experimental stations, Gansu, China (34°58′N, 104°23′E, 2,074 m altitude), 2023.
It is not clear how authors might achieve the aims with such different factors going on.
Thanks for your positive comments. That's a very good question. In the manuscript, to analyze the expression patterns of all BXs biosynthetic gene cluster (BGC) members in various biotic/abiotic-induced tissues of maize, three independent RNA-sequencing (RNA-Seq) analyses were performed in mesocotyls and coleoptiles of Zheng58 seedlings under four light spectral quality (Experiment 1); in young leaves of Ji853 and N192 seedlings with or without 2 μM 24-epibrassinolide (EBR) at 25 and 15 °C conditions (Experiment 2), in whorl leaves of F1227 and M0800 plants to feeding and non-feeding conditions by Asian corn borer (ACB) (Experiment 3). After filtering, using the HISAT v.2.2.1 (https://daehwankimlab.github.io/hisat2/; accessed on 18 December 2023), the clean sequence reads of above three RNA-Seq were aligned to the Zea mays B73_V4 reference genome, respectively. Subsequently using the FPKM values to estimate the genes expression of 26 BGC gene members via the Cufflinks v.2.2.1 [55], and which were visualized using the TBtools software (https://github.com/CJ-Chen/TBtools/releases; accessed on 25 December 2023).
The maize materials including Zheng58, Ji853, N192, F1227, and M0800, they have complex genetic background (Niu, Y.N.; Zhao, X.Q.; Chao, W.; Lu, P.N.; Bai, X.D.; Mao, T.T. Genetic variation, DIMBOA accumulation, and candidate gene identification in maize multiple insect-resistance. Int. J. Mol. Sci. 2023, 24, 2138.), thereby there may be significant differences on the expression patterns of the 26 BGC gene members in various tissues of above five maize materials under low temperature, light spectral quality, 24-epibrassinolide (EBR) induction, and Asian corn borer (ACB) infestation.
To further identification core conserved BGC gene members that were stably expressed in various maize materials, tissues, and environmental stresses. For above each independent RNA-Seq, the TPM values of these BGC gene members based on the normalized scale method were calculated [58]. Meanwhile their significantly differential expression in all comparisons were estimated using a P-value < 0.05 and |log2 FC| > 1 criterion. Firstly, the common BGC gene members were identified that showed both TPM value > 2 in single sample, and |log2 FC| > 1 and P-value < 0.05 in single comparison in an independent RNA-Seq. Then the core conserved BGC gene members with high, specific, and significant expression were identified among three common BGC gene members of all RNA-Seq by the VENNY 2.1 (https://bioinfogp.cnb.csic.es/tools/venny/index.html; accessed on 29 December 2023).
At the same time, to validate the expression patterns of core conserved DEGs of BGC, six core conserved DEGs of BGC gene members were randomly selected for qRT-PCR and DIMBOA contents were quantified in young leaves of 14-day old F1227 seedlings in response to light, temperature, and wound (mimicing ACB feeding symptom). The qRT-PCR analysis showed that these core conserved BGC gene members were significantly activated under both darkness and wound treatments, whereas significantly decreased under low-temperature treatment, compared to the relative controls (Figure 8A). Surprisingly, DIMBOA content significantly increased by 8.3 and 4.0% under darkness and wound treatments, compared to control treatment in young leaves of F1227 seedlings, respectively (Figure 8B). However, low-temperature treated leaves showed the opposite trend, i.e., with a significant decrease (46.8%) in DIMBOA level (Figure 8B). Further Pearson correlation analysis showed that the expression levels of the six core conserved BGC gene members were significantly (p < 0.05) positive correlation to DIMBOA accumulation in young leaves of 14-day old F1227 seedlings in response to four treatments (Figure 8C). Therefore, light, temperature, and wound significantly impacted the expression of corresponding core-conserved BGC gene members, suggesting that they might play a role in response to these stresses, via the BXs accumulation, especially obvious changes of DIMBOA contents.
Thank you for your consideration.
- Is the data new? As it is written, the raw data seems to come from previous studies. This is not clear.
Thanks for your positive comments. That's a very good question. For the three independent RNA-sequencing (RNA-Seq) analyses in mesocotyls and coleoptiles of Zheng58 seedlings under four light spectral quality (Experiment 1); in young leaves of Ji853 and N192 seedlings with or without 2 μM 24-epibrassinolide (EBR) at 25 and 15 °C conditions (Experiment 2), in whorl leaves of F1227 and M0800 plants to feeding and non-feeding conditions by Asian corn borer (ACB) (Experiment 3). The raw data of Experiment 1 were published in our previous study (Zhao, X.Q.; Niu, Y.N.; Hossain, Z.; Zhao, B.Y.; Bai, X.D.; Mao, T.T. New insights into light spectral quality inhibits the plasticity elongation of maize mesocotyl and coleoptile during seed germination. Front. Plant Sci. 2023, 14, 1152399.). However, the raw data of Experiment 2 and Experiment 3 were new in the manuscript.
Therefore, to describe the raw data of above three independent RNA-Seq more clearly, as suggested, we have revised the corresponding contents, namely: “The published raw data of this RNA-Seq was deposited to NCBI BioProject database (accession number PRJNA917698).” “The new raw data of this RNA-Seq was deposited to NCBI BioProject database (accession number PRJNA1096968).” “The new raw data of this RNA-Seq was deposited to NCBI BioProject database (accession number PRJNA1121054).”
We then have re-submitted the manuscript.
Thank you for your consideration.
- were rt-PCR analyses done for all genotypes? were the same samples used to analyze gene expression?
Thanks for your positive comments. That's a very good question. To validate the expression patterns of core conserved DEGs of BGC, six core conserved DEGs of BGC gene members were randomly selected for qRT-PCR and DIMBOA contents were quantified in young leaves of 14-day old F1227 seedlings in response to light, temperature, and wound (mimicing ACB feeding symptom).
For qRT-PCR analysis of core conserved DEGs of BGC gene members. The 20 soaked seeds of F1227 with ddH2O, were sown in plastic boxes, and cultured for two weeks under four treatments in corresponding growth chambers, including control treatment (CK; seedlings growing in normal environment; 22±0.5°C, 300 μM m-2 s-1 white light, 12 h photoperiod), darkness treatment (Dark; etiolated seedlings growing in darkness, 22±0.5°C), low-temperature treatment (LT; seedings growing at 22±0.5°C for seven days and then culturing at 10±0.5°C for seven days; 300 μM m-2 s-1 white light, 12 h photoperiod), and wound treatment (Wound; three leaf wounds of 3×3 mm were made with a scalpel on 13-day old seedling for 24 h [59]; 22±0.5°C, 300 μM m-2 s-1 white light, 12 h photoperiod). Next, the total RNA was extracted with TRIZOL reagent (TIANGEN, Beijing, China) of young leaves of F1227 seedlings under four above treatments, and then 0.5 μg RNA was reverse-transcribed to produce first-stand cDNA using PrimeScriptTM 1st stand cDNA synthesis Kit (TaKaRa, Japan). The Primers of six random core conserved BGC gene members were designed and synthetized [10,20] (Table S6). Subsequently the qRT-PCR analysis was performed using LightCycler480II fluorescent quantitative PCR instrument (Roche, Germany). The relative expression level with three biological replicates was calculated by the 2−ΔΔCT method, with Zm00001d010159 (Actin 1) as an internal reference gene [60].
For DIMBOA content determination. The DIMBOA content in young leaves of F1227 seedlings under above treatments was determined according to our previous report [29]. The freeze-dried leaves (0.2 g) were homogenized into screw-capped 10 mL centrifuge tubes and 5 mL methanol-methanoic acid solution (0.01%, v/v) was added to corresponding tube. The tubes were rotated and placed in darkness for 12 h. and then centrifuged at 12,000 rpm (Centrifuge 5425/5425 R; Eppendorf, Germany) for 20 min at 4°C. The 600 μL supernatants were then slowly passed filtration column for HPLC-MS analysis. DIMBOA Standard (CAS: 15893-52-4) was used to optimize the mass spectrometric parameters and fragment spectra.
In addition, we have also re-added the corresponding contents, namely: “4.7. Statistical Analyses For the qRT-PCR expression levels of six core conserved BGC gene members and DIMBOA content in young leaves of F1227 seedlings under above treatments, their ANOVA was performed using the IBM-SPSS Statistics v.19.0 (https://www.Ibm.com/products/spss-statistics, accessed on 23 May 2024), and their interactive ring Pearson correlation diagram was drew using the Genescloud tool (https://www.genescloud.cn, accessed on 24 May 2024).”
We then have re-submitted the manuscript.
Thank you for your consideration.
- Overall, there is a wide discrepancy between methods/results and what is written in the abstract.
Thanks for your positive comments. As suggested, we have carefully revised and improved the Abstract section, Results section, and Materials and methods section of the manuscript, to make the full text consistent. We then re-submitted the manuscript.
Thank you for your consideration.
Figures: the text in several figures cannot be followed.
Thanks for your positive comments. As suggested, we have carefully checked the all figures and then improved the corresponding Figures, including Figure 3 and Figure 8. We have also re-added the Figure S1 in Supplementary Figure S1. Moreover, we have checked the position of each Figure in the manuscript.
We then re-submitted the manuscript.
Thank you for your consideration.
"Supplementary Materials: The following supporting information can be downloaded from https://www.mdpi.com/article/. References [13,20,58] are cited in the supplementary Table S6." - the link does not work. I did not have access to any supporting information files.
Thanks for your positive comments. As suggested, we have re-uploaded the supplementary materials in the submission system of the International Journal of Molecular Sciences. These Supplementary materials were:
Table S1. The annotations of the 26 identified benzoxazinoids (BXs) biosynthetic gene cluster (BGC) gene members in maize.
Table S2. The physiochemical parameters and subcellular location of the 26 identified benzoxazinoids (BXs) biosynthetic gene cluster (BGC) gene members in maize.
Table S3. The transcripts per million (TPM) values of the 26 identified benzoxazinoids (BXs) biosynthetic gene cluster (BGC) gene members in coleoptiles and messocotyls of Zheng58 for germination ten days in red, blue, white light and darkness by RNA-sequencing.
Table S4. The transcripts per million (TPM) values of the 26 identified benzoxazinoids (BXs) biosynthetic gene cluster (BGC) gene members in young leaves of Ji853 and N192 seedlings at V3 stage with or without 2 μM 24-epibrassinolide (EBR) at 25 and 15 °C.
Table S5. The transcripts per million (TPM) values of the 26 identified benzoxazinoids (BXs) biosynthetic gene cluster (BGC) gene members in whorl leaves of F1227 and M0800 plants at VT stage to feeding and non-feeding by Asian Corn borer (ACB).
Table S6. Quantitative real-time (qRT-PCR) primers used to expression levels of eight benzoxazinoids (BXs) biosynthetic gene cluster (BGC) gene members.
Figure S1. Weather data in all maize life stages at the Longxi experimental stations, Gansu, China (34°58′N, 104°23′E, 2,074 m altitude), 2023.
Thank you for your consideration.
Proofreading comments: the text has several typos and grammatical mistakes.
Thanks for your positive comments. As suggested, we have carefully checked the typos and grammatical mistakes of the manuscript, and revised and improved the manuscript using the “Track Changes” function. We then re-submitted the manuscript.
We then re-submitted the manuscript.
Thank you for your consideration.
Open Review: I would like to sign my review report.
Thanks for your positive comments.
Thank you for your consideration.
Quality of English Language: Moderate editing of English language required.
Thanks for your positive comments. As suggested, under the guidance of Professor Xiquan Gao (who is a famous expert in maize and has published several influential articles, his English is very idiomatic; https://mp.weixin.qq.com/s/6Hbtdq50MbR4vh4qh6fBFg), we have further revised and improved the English language of the manuscript. We then re-submitted the manuscript.
Thank you for your consideration.
Does the introduction provide sufficient background and include all relevant references? Yes.
Thanks for your positive comments.
Thank you for your consideration.
Is the research design appropriate? Must be improved.
Thanks for your positive comments. As suggested, we have improved the experimental designs in detail. We then re-submitted the manuscript.
Thank you for your consideration.
Are the methods adequately described? Must be improved.
Thanks for your positive comments. As suggested, we have improved the experimental methods in detail. We then re-submitted the manuscript.
Thank you for your consideration.
Are the results clearly presented? Must be improved.
Thanks for your positive comments. As suggested, we have revised and improved the Results section of the manuscript. We then re-submitted the manuscript.
Thank you for your consideration.
Are the conclusions supported by the results? Can be improved.
Thanks for your positive comments. As suggested, we have revised and improved the Conclusion section of the manuscript. We then re-submitted the manuscript.
Thank you for your consideration.
Sincerely,
Xiaoqiang Zhao professor
State Key Laboratory of Aridland Crop Science, Gansu Agricultural University
E-mail: zhaoxq3324@163.com

Reviewer 2 Report
Comments and Suggestions for Authors> Title of this manuscript is too lengthy, need to make it brief and understanding.
> Abstract written in good format and style with the Rythm but authors need to clearer the conclusion and future recommendations for reviewers.
> Keyword should not belong to title or abstract part. Need to rephrase them.
> In Figure 2b red dotted line showing the same trend as the top of each bar, I preferred to change the bar length to for frequency. So, reader can easily judge the graphs.
>In Figure 3A Its better to remove the UTR from the Gene structure to make it more align to each gene.
> In Fig 3B another way if authors arranged the Motif analysis with the Gene Structure will be more understandable for reviewers.
> Figure 4, 5 and 6 the bars for log2 can't see here, as in general I can't what that value belongs to and which part it linked with.
> In Figure 8 my suggestion is to move the DIMBOA graphs on Side and move the relative expression graphs in two rows with 3 columns.
> Metrial and method are well written with everything clear.
Author Response

(The authors gave the same response as above.)

Reviewer 3 Report
Comments and Suggestions for Authors
In this manuscript, the authors aim to:
"(i) identify BXs biosynthetic gene cluster (BGC) gene members in maize, and comprehensively analyze their phylogenetic relationships, sequence features, chromosome distributions, and subcellular localization;
(ii) analyze the expression patterns of all BGC gene members in various biotic/abiotic-induced tissues of maize via RNA-sequencing (RNA-Seq);
(iii) verify the relative expression patterns of the core conserved BGC genes in response to different biotic/abiotic stresses by quantitative real-time PCR (qRT-PCR) and HPLC."
Overall, the manuscript has a great potential. However, methods are not clear and the authors seem to have used different tissues to achieve these aims. Without further clarification, these raise several concerns. I have some doubts:
- Methods lack some details. For instance, ". Then, the coleoptiles and mesocotyls of Zheng58 seedlings under four treatments, with three biological replicates, in total 24 samples were used for RNA-Seq [54]." - which treatments? Any details about Zheng58? Why was it used?
For Ji853 (chilling-sensitive) and N192 (chilling-tolerant) , the authors wrote ". Overall, the young leaves of both maize seedlings under four treatments, with three biological replicates, in total 24 samples were used for RNA- Seq." Here leaves were rather used? Why? Again, which treatments?
For F1227 (insect-resistant) and M0800 (insect-susceptible), the authors used "the whorl leaves of F1227 and M0800 plants under two treatments, with three biological replicates, in total 12 samples were used for RNA- Seq."
It is not clear how authors might achieve the aims with such different factors going on.
- Is the data new? As it is written, the raw data seems to come from previous studies. This is not clear.
- were rt-PCR analyses done for all genotypes? were the same samples used to analyze gene expression?
- Overall, there is a wide discrepancy between methods/results and what is written in the abstract.
Figures: the text in several figures cannot be followed.
"Supplementary Materials: The following supporting information can be downloaded from https://www.mdpi.com/article/. References [13,20,58] are cited in the supplementary Table S6." - the link does not work. I did not have access to any supporting information files.
Proofreading comments: the text has several typos and grammatical mistakes.
Comments on the Quality of English Language
see above
Author Response

(The authors gave the same response as above.)

Round 2
Reviewer 3 Report
Comments and Suggestions for Authors
The authors have erased all previous concerns/issues.